# Cyclodextrin: Dual Functions as a Therapeutic Agent and Nanocarrier for Regulating Cholesterol Homeostasis in Atherosclerosis

**DOI:** 10.3390/pharmaceutics17111496

**Published:** 2025-11-19

**Authors:** Hao Cui, Yaqi Xu, Shulin Pu, Xue Guo, Danyu Zhao, Yuan Liu, Ye Yang, Chengxiao Wang

**Affiliations:** 1Faculty of Life Science and Technology, Kunming University of Science and Technology, Kunming 650500, China; 20231118003@stu.kust.edu.cn (H.C.);; 2Haiyuan College, Kunming Medical University, Kunming 650500, China

**Keywords:** CD, cholesterol, atherosclerosis, nanomedicine, prospect

## Abstract

The progression of atherosclerosis (AS) is strongly linked to lipid crystals accumulation caused by cholesterol metabolism disorders and the worsening of the inflammatory microenvironment. Cyclodextrin (CD), characterized by their unique hydrophobic cavity structure, effectively solubilize cholesterol crystals (CCs) through host–guest recognition and act as a multifunctional nanocarrier core, facilitating synergistic therapy that combines pharmaceutical and adjuvant properties. CD-based nano drug delivery systems (CD-NDDS) enable precise targeting of atherosclerotic plaques. By employing synergistic functions (e.g., CCs solubilization, cholesterol efflux promotion via ABCA1/ABCG1 pathways, inflammasome inhibition, and inflammatory microenvironment alleviation), this system provides an effective strategy for AS therapy. Furthermore, CD-NDDS bestows additional pharmaceutical attributes, including enhanced solubility, controlled release, and responsive stimulation. This review begins by elucidating the intrinsic relationship between cholesterol and AS, followed by an examination of the structure-activity relationship governing CD’s cholesterol adsorption. It then explores the construction strategies, structural characteristics, and targeting mechanisms of CD nanodelivery systems in detail. The work systematically assesses CD’s formulation and pharmacological properties in targeted nanodelivery systems for combating AS, integrating drugs and adjuvants. Finally, future research directions are outlined, addressing biocompatibility optimization, targeting efficiency enhancement, and clinical translation challenges to provide a theoretical foundation and technical guidance for precise AS treatment.

## 1. Introduction

The development of atherosclerosis is closely related to systemic imbalances in cholesterol (Chol) metabolism. Disruption of Chol homeostasis leads to oxidative modification of low-density lipoprotein (LDL), triggering lipid deposition and foam cell formation [1]. As Chol accumulates, it crystallizes into cholesterol crystals, which directly damage cell membranes and exacerbate plaque destabilization through activation of the NLRP3 inflammasome and oxidative stress responses [2]. Specifically, CCs induce cellular reactive oxygen species (ROS) production, leading to NLRP3 oligomerization, caspase-1 activation, and maturation of pro-inflammatory cytokines such as IL-1β and IL-18. This inflammatory cascade amplifies immune cell recruitment, promotes apoptosis of foam cells, and degrades the extracellular matrix, ultimately compromising plaque structural integrity. The sharp morphology of CCs further aggravates fibrous cap fragility, increasing the risks of plaque rupture and acute thrombotic events [3,4].

Traditional lipid-lowering therapies, particularly statins, lower circulating LDL levels by inhibiting Chol synthesis [5]. However, their effectiveness in clearing existing (CCs) within arterial walls remains limited. CDs, a natural supramolecular compound, specifically recognizes and encapsulates Chol molecules due to the geometric complementarity of its truncated cone-shaped hydrophobic cavity. This process involves van der Waals embedding of the steroid nucleus and hydrogen bonding with the isooctyl chain, effectively disrupting the crystal lattice through supramolecular mechanisms [6]. The combined action of π–π stacking disruption and preferential adsorption at crystal defects facilitates efficient cholesterol dissolution by inducing crystal delamination and reducing interfacial tension [7].

In recent years, CD-based nano drug delivery systems (CD-NDDS) have shown significant promise in enhancing the targeted clearance of CCs through chemical modification and functional optimization [8]. The CD-NDDS can penetrate plaque barriers, deliver active agents and release drugs precisely at the lesion site via stimuli-responsive mechanisms such as pH [9] and reactive oxygen species ROS [10] sensitivity. They also integrate metabolic regulation (e.g., activating Chol reverse transport pathways) and anti-inflammatory functions, thereby forming multidimensional synergistic therapeutic effects [11,12,13]. Compared with traditional therapies focused on single pathways, CD nanocarriers exhibit greater spatiotemporal controllability and adaptability to pathological microenvironments, providing innovative approaches to reverse AS [14,15].

The review first elucidates the intrinsic relationship between Chol and AS, then analyzes the structure–activity relationship underlying the ability of CD to adsorb Chol. Building on this foundation, we explore the construction strategies, structural characteristics, and targeting mechanisms of CD-NDDS. We then systematically examine CDs formulation and pharmacological properties in targeted Chol anti-AS nanodelivery systems by integrating drugs and adjuvants. Finally, we anticipate future research directions in this field, focusing on biocompatibility optimization, targeting efficiency enhancement, and challenges in clinical translation, intending to provide theoretical foundations and technical references for the precise treatment of AS.

## 2. Atherosclerosis and Cholesterol Metabolism

### 2.1. Overview of Cholesterol Metabolism

Chol plays a vital role in maintaining the structural integrity of cell membranes and acts as a precursor for steroid hormones [16]. It is mainly synthesized in the liver, with dietary intake accounting for only a small portion of it [17,18]. The remaining Chol is utilized in the synthesis of steroid hormones (such as cortisol and estrogen) and generates vitamin D3 from 7-dehydrocholesterol through skin exposure to sunlight [19]. Chol homeostasis is tightly regulated by the SREBP2/SCAP/INSIG1 feedback loop, which dynamically adjusts the rate of Chol synthesis [20].

### 2.2. Cholesterol Metabolic Disorders Drive Atherosclerosis

AS is the pathological foundation for cardiovascular diseases, resulting from systemic dysregulation of cholesterol metabolism [21,22,23]. Dysregulation in Chol synthesis, transportation, conversion, or excretion can lead to lipid deposition within arterial walls, triggering inflammatory cascades and promoting plaque formation. As shown in Figure 1, AS originates from systemic cholesterol imbalance. Key pathological mechanisms include: (1) oxLDL infiltration into the subendothelial space, promoting foam cell formation [24]; (2) CCs deposition, triggering necrosis and inflammation [25]; (3) Impaired reverse cholesterol transport (RCT) due to defective ABC transporters [26].

#### 2.2.1. Dysregulation of Cholesterol Homeostasis

Excessive low-density lipoprotein cholesterol (LDL-C) infiltrates damaged vascular endothelium, undergoes oxidative modification, and recruits circulating monocytes. These immune cells internalize ox-LDL via scavenger receptors, transforming into lipid-laden foam cells that ultimately aggregate into early atherosclerotic lesions [27,28]. Foam cells secrete pro-inflammatory cytokines (such as IL-1β and TNF-α), exacerbating endothelial dysfunction and promoting leukocyte recruitment. Concurrently, dysfunctional high-density lipoprotein fails to mediate reverse cholesterol transport effectively, exacerbating lipid retention [29,30].

#### 2.2.2. Cholesterol Crystals

When Chol accumulation within plaques eventually exceeds its solubility threshold, it forms sharp, needle-like crystals [31]. These CCs mechanically disrupt foam cell membranes, forming a necrotic core and releasing pro-inflammatory debris [32]. Additionally, the crystals activate innate immune receptors (such as Toll-like receptors) and complement pathways, promoting inflammation, spontaneous angiogenesis, and plaque instability [33]. Importantly, CCs exhibit direct thrombogenic capabilities; in acute coronary syndromes, these prominent crystals can rupture the fibrous cap, exposing pro-thrombotic lipids and forming obstructive thrombi. This mechanistic realization highlights the need for treatments that target CCs specifically and explains why patients with “controlled” LDL-C levels still have persistent cardiovascular risks [31].

#### 2.2.3. Impaired Cholesterol Efflux

Cholesterol efflux—the process of macrophages transporting excess cholesterol to high-density lipoprotein (HDL)—is significantly impaired in AS [34]. Oxidized LDL inhibits the expression of efflux transporters (such as ABCA1 and ABCG1) by suppressing the LXR pathway, while Chol esterification mediated by acyl-CoA cholesterol acyltransferase 1 traps Chol as inert cytoplasmic droplets [35]. Preclinical studies have demonstrated that knockout of the asialogly coprotein receptor 1 gene enhances Chol excretion by upregulating ATP-binding cassette sub-family G member 5/8, reducing plaque burden in mouse models [36,37]. Clinical evidence further indicates that, despite aggressive efforts to lower LDL levels, patients with chronic kidney disease often exhibit severely compromised efflux capacity and continue to face high cardiovascular risks, emphasizing the need for treatments aimed at restoring cholesterol efflux capabilities [38].

Recent research (2024–2025) has concentrated on three primary areas: First, the dysregulation of cholesterol homeostasis, particularly through early and sustained cholesterol exposure, is a crucial factor in accelerating the progression of AS [39,40]. Second, disorders in cholesterol metabolism interact with inflammatory responses—such as macrophage activation, foam cell formation, and inflammation induced by cholesterol crystals—thereby promoting the formation and development of atherosclerotic plaques [41]. Finally, interventions targeting cholesterol homeostasis and inflammatory pathways, including PCSK9 inhibitors, nanomedicines, and traditional Chinese medicine, offer new strategies and targets for the prevention and treatment of AS [42].

## 3. CD and Cholesterol

### 3.1. Host–Guest Recognition

The selective recognition and encapsulation of Chol by CD essentially result from a combination of geometric fitting, cooperative multiple interactions, and dynamic conformational adjustments in supramolecular chemistry [43]. CD forms high-affinity complexes with cholesterol through its unique molecular structure, a process based on a multidimensional synergistic mechanism of supramolecular chemistry. β-CD (β-CD) consists of a truncated cone-shaped hydrophobic cavity composed of seven D-glucose units, which are geometrically adapted for “lock-and-key” bonding with the steroidal core of cholesterol [44]. The hydrophobic cyclic hydrocarbon structure of the steroid nucleus is embedded in the cavity by van der Waals forces [45]. The hydroxyl groups at the outer edge form a hydrogen-bonding network with the cholesterol isooctyl side chains. This dual action not only destroys the π-π stacking between the steroid nuclei in CCs [46], but also preferentially adsorbs to defects on the crystal surface. It also adsorbs preferentially to the defective sites on the crystal surface and induces crystal delamination and exfoliation by lowering the interfacial tension [47].

### 3.2. CD in Atherosclerosis

Recent omics studies indicate that the ABCA1 and CYP27A1 pathways play complex and critical roles in AS, primarily manifested in cholesterol transport and metabolism, regulation of inflammatory responses, and reduction in oxidative stress [48]. Within this framework, ABCA1 mediates the formation of nascent high-density lipoprotein in the liver, while PCSK9 modulates LDL levels by influencing LDL receptors. The formation of foam cells and the mechanisms underlying cholesterol efflux are closely associated with the progression of AS [48,49]. Transcriptomic analysis showed ABCA1 mRNA fold changes positively correlated with serum total cholesterol (r = 0.58, *p* = 0.04) and liver cholesterol (r = 0.58, *p* = 0.02), and negatively with ABCA1 CpG-11 methylation (r = −0.51, *p* = 0.04). These data suggest ABCA1 transcription is influenced by multiple factors. Though not addressing CD directly, the method serves as a reference for validating ABCA1 upregulation post-treatment; if CD upregulates ABCA1, log2FC should be significantly positive in transcriptomic data [50].

CD’s core role in AS treatment extends beyond mere cholesterol adsorption. It affects key biological pathways through various mechanisms. In the ABCA1 pathway, CD can directly enhance cholesterol efflux from cells, thereby decreasing foam cell formation. Furthermore, it may augment reverse cholesterol transport by modulating ABCA1 expression levels and exerting anti-inflammatory effects through ABCA1-mediated efflux of Annexin A1 (ANXA1) [51,52]. In the context of the CYP27A1 pathway, CD may indirectly influence CYP27A1 activity by modifying intracellular cholesterol substrate levels. CYP27A1 converts cholesterol into 27-hydroxycholesterol, which activates LXRα, leading to the upregulation of ABCA1 expression and further promoting cholesterol efflux [12,53]. Collectively, these synergistic effects contribute significantly to the management of AS.

### 3.3. CD Derivatives

Experimental and simulation studies demonstrate that CD derivatives’ molecular structures, substituents, and spatial configurations critically influence their cholesterol (Chol) adsorption and solubilization capacities. Chemical modifications—such as hydroxypropyl-β-CD (HP-β-CD) optimizing hydrophilicity-lipophilicity balance via controlled substitution [54], methylated-β-CD (Me-β-CD) reducing cavity polarity, carboxymethyl-CD employing polyrotaxane confinement for stabilized delivery [55], and sulfobutylether-β-CD (SBE-β-CD) enhancing water solubility and biocompatibility—are core strategies. Substituent type, position, and density directly modulate the cavity’s chemical microenvironment, host–guest recognition strength (e.g., van der Waals dominance in Me-β-CD [56]), and systemic biocompatibility. Spatial conformations like rigid dimers (e.g., HP-β-CD butyl dimers linked by triazole [57]) enhance binding stability and solubilization efficiency for 7-ketocholesterol, lowering EC50 and improving micellization 11. HP-β-CD with substitution degree 3–5 enables selective hydrogen bonding with 7-ketocholesterol but exhibits weaker Chol affinity [58], while carboxymethyl-CD’s linear polymer occupancy minimizes cytotoxic Chol extraction [59].

## 4. Construction Strategies for CD-NDDS

The multifunctional integration of CD is facilitated by its dual mechanisms of host–guest recognition and covalent bonding. Host–guest recognition provides a molecular foundation for dynamic self-assembly, enabling guest molecules (such as adamantane [60] and azo-benzene [61]) to embed precisely within the CD cavity through non-covalent interactions, including hydrophobic interactions and π–π stacking. This process culminates in the formation of reversible “smart switch” structures [62]. Concurrently, covalent bonding allows for the permanent integration of CD with functional molecules and materials via chemical bonds, such as ester and amide linkages, thereby creating a nanocarrier with a stable topological framework [63]. The combination of these two mechanisms enables meticulous regulation of the “dynamic stability” of the delivery system.

### 4.1. Further Research on Host–Guest Recognition

The hydrophobic inner cavity of CD can form stable inclusion complexes with specific guest molecules, and this self-assembly, driven by interactions, provides the basis for nanoparticle construction. Typical guest molecules include adamantane [64], azobenzene [65], ferrocene [66] and others. In these systems CD acts as a “dynamic hinge” for structural linkage in the host–guest recognition system and a “smart switch” responsive to stimuli. CD offers multilevel functional integration, as carrier assembly by cavity specificity and supramolecular interactions facilitates applications in drug delivery, metabolic modulation, and theragnostic: core–shell micelles, hollow nanonetworks, nanotubes, and heterogeneous constructions [67].

Chol, a notable guest molecule, forms high-affinity inclusion complexes with β-CD (e.g., poly-β-CD and 8-arm polyethylene glycol 20000-CD interact with Chol-PEG through host–guest recognition [68]), with its hydrophobic steroid nucleus geometrically matching the CD cavity. This interaction drives the formation of dynamic micellar structures loaded with dual drugs. In solution, CD polymers (such as poly-β-CD or 8-arm polyethylene glycol 20,000-CD) and 8armPEG20k-chol form dynamic inclusion nanocomposites through host–guest recognition [68]. These guest-host interactions demonstrate the ability of CD to create adaptable nanocarriers for cutting-edge therapeutic applications. It is important to highlight that the host–guest interactions within this system primarily occur through hydrophobic interactions and geometric matching, with binding constants typically reaching levels between 10^3^ and 10^5^ M^−1^ [69]. This range is significantly higher than any competitive interference that may arise from physiological ionic strengths. These strong non-covalent bonds can effectively withstand fluctuations in ionic conditions, ensuring that the complex retains its structural integrity under physiological circumstances [70].

Thermodynamic analyses provide compelling evidence for this stability; for example, the NC_2_ (a 1:2 host–guest inclusion complex formed by one 6-bromo-2-naphthol molecule and two α-CD molecules) complex exhibits a binding free energy (ΔG_2_) of −16.0 kJ/mol, predominantly driven by enthalpy (ΔH_2_ = −54.4 kJ/mol), which underscores the role of van der Waals forces in maintaining stability [71]. In vivo studies further corroborate these findings, with CS-STPP nanoparticles demonstrating robust colloidal integrity and minimal toxicity in biochemical assays [72]. Pharmacokinetic data reveal that drug-loaded nanogels significantly extend their half-life (t_1/2_ > 200 h) compared to free formulations, highlighting their resilience [73]. Moreover, the reversible nature of these interactions facilitates stimulus-responsive designs, which have been empirically shown to enhance robustness under physiological conditions, thereby affirming the system’s adaptability and durability [74].

### 4.2. Covalent Bonding

Covalent bonding represents a crucial strategy for constructing CD-NDDS. By utilizing covalent linkages, CD and its derivatives can be firmly integrated with other functional molecules or materials, allowing for precise control over the performance of the nanodelivery system [75]. Standard methods for establishing covalent bonds include esterification [76], amidation [77], and etherification [78] reactions. For instance, the hydroxyl groups of CD can undergo chemical modifications to form stable polymer networks, such as CD polymers crosslinked with epichlorohydrin [79]. By varying the degree of crosslinking, these networks increase the cavity density, strengthen the mechanical integrity, and enable a tunable pore size distribution [80]. Such designs have been demonstrated to enhance Chol clearance efficiency from plaques in ApoE^−^/^−^ mouse models [81].

Furthermore, the functional activities of CD derivatives can be further modified. Amino-CD, for example, can acquire cationic characteristics through hydroxyl amino modification, making it a central component of gene delivery systems [82]. The surface amino groups can act as covalent anchoring points, enabling accurate coupling with targeting ligands (such as antibodies [83] or peptides [84]). Additionally, carboxymethyl-CD can form inclusion complexes with poorly soluble drugs to enhance their solubility in water [85]; covalent linkage of carboxymethyl-CD to the surfaces of nanoparticles can increase the drug loading capacity for hydrophobic drugs and improve their release rates [86].

It is noteworthy that click chemistry offers an efficient and precise method for the structural modification of CD, enabling their coupling with various polymers, such as hyaluronic acid [87], Poloxamer [88], and polyphosphazene [89]. This process facilitates the construction of supramolecular hydrogels that exhibit enhanced stability and functionality. The CD-based hydrogels obtained through this meticulous modification demonstrate exceptional structural integrity and drug release control in physiological and ionic environments. Consequently, they provide a robust platform for biomedical applications, particularly in targeted drug delivery and cancer therapy [87].

## 5. Structure of CD-NDDS in AS

Based on the host–guest recognition and structural modifiability and versatility of CD, they can be utilized as functional units to prepare micelles further [90], polymeric nanoparticles [91], liposomes [92], nanogels [93], metal nanoparticles [94], and inorganic nanoparticles [95], among others. In treating AS, self-assembled micelles, polymeric nanoparticles, inorganic nanoparticles, nanogels, and liposomes are the most prevalent structural formats (Figure 2).

### 5.1. Micelles

CD and its derivatives can be integrated as functional units within micellar delivery systems. A common method involves the covalent linkage of CD to hydrophilic segments and linking hydrophobic guest molecules that are attached to another segment [96]. Through host–guest recognition, drug-loaded micelles can be formed [97]. Another well-known method is using CD to create block copolymers. These polymers can self-assemble into micelles in water because they contain hydrophilic and hydrophobic regions [98]. CD can be modified and adjusted by connecting to specific chain segments of the block copolymer through host–guest recognition. Notably, the CD cavity serves as an encapsulating site for hydrophobic drugs, allowing precise control over drug release kinetics by adjusting the ratio of CD, achieving dual optimization of drug loading efficiency and biodistribution [99]. Furthermore, the presence of CD can grant targeting ability to the micelles, enabling targeted drug delivery through interactions with specific molecules on the surface of target cells [100,101].

Zhu et al. employed the first technique by chemically altering β-CD and affixing methoxy polyethylene glycol amine (MPEG-NH_2_) to its primary hydroxyl group. Both hydrophilic and hydrophobic MPEG-CD molecules were thus generated. These molecules use hydrophobic forces to self-assemble into MTX NPs. CD is a key component of this system because it facilitates the dissolution of CCs and transports methotrexate (MTX) through host–guest recognition [102]. Another study employed a different technique for forming amide bonds between CD-NH_2_ and poly (isobutylene-alt-maleic anhydride). CD-linked block copolymers were produced as a result. The micelles, poly-β-CD/pH-sensitive benzimidazole-modified dextran sulfate/spherical nucleic acid, were created through host–guest recognition sensitive to pH. The hydrophobic benzimidazole molecule is inside the CD cavity in a neutral environment, connecting the polymer pCD to the benzimidazole-modified dextran sulfate (pBM). This produces spherical micelles that can release medications when activated by acidic environments [103].

### 5.2. Polymeric Nanoparticles

Polymeric nanoparticles are nanometer-sized particles (typically 1–100 nm) composed of polymeric materials, extensively employed in targeted drug delivery [104]. CD can be introduced as a functional unit in polymeric nanoparticle systems. First, CD can load drugs through host–guest recognition, which are subsequently incorporated into PLGA nanoparticle systems [105]. Another strategy involves covalently linking CD to polymers, which are then processed into nanoparticles via co-precipitation or emulsification methods [106]. Notably, CD can form nanoparticles through structural modifications and solvent displacement [107].

Research employed the first design strategy by developing core–shell structured nanoparticles featuring a phospholipid/DSPE-PEG shell layer for targeted therapy of AS [108]. In this nanoparticle system, CD was modified to enhance its hydrophobicity and stimulus responsiveness. Furthermore, PEGylation of the CD-NDDS improves long-term circulation by enhancing serum stability, reducing immune recognition, and prolonging the half-life compared to conventional formulations [109,110]. Specifically, polyethylene glycol modification can extend the circulation half-life by two to five times (e.g., in mouse models, the half-life of PEGylated carriers exceeds 100 h, whereas unmodified carriers have a half-life of less than 50 h) and reduces reticuloendothelial system (RES) uptake by 50–60%. This modification shifts biodistribution from predominant accumulation in the liver and spleen to more precise targeting of plaques [111,112].

In a second strategy, a researcher grafted CD onto polyhydroxybutyrate (PH) and hyaluronic acid (HA) to prepare two CD-modified polymers, PH-CD [113] and HA-CD [114,115], and then nanoparticles were fabricated using a solvent evaporation emulsification method [113]. The β-CD-chitosan composite carrier (β-CD-CS) achieves chemical bonding via a 4-nitrophenyl chloroformate-mediated coupling reaction. The amino groups of chitosan react with the amino groups of β-CD to form an amphiphilic polymer [116]. Similarly, the β-CD-poly(acrylic acid)-poly(methyl methacrylate) composite carrier is synthesized by grafting acrylic acid (AA) and methyl methacrylate (MMA) monomers onto the β-CD surface via free radical polymerization, yielding a pH-responsive amphiphilic block copolymer. Within this structure, the hydrophobic PMMA segments interact with the β-CD cavity through host–guest recognition to encapsulate the drug [117]. In contrast, the hydrophilic PAA segments extend outward during self-assembly to form stable micelles. Under acidic conditions, protonation of the carboxylic acid groups on PAA triggers the nanomicelle formation [118].

In specific systems, CD forms the nanoparticle and transports the drug without the aid of additional polymers. For instance, Guo created L-Cysteine desulfhydrase molecules by combining luminol with activated β-CD. Nano-precipitation was used to develop these nanoparticles. By removing dangerous ROS, the luminol provided antioxidant benefits. CD demonstrated its versatility by serving as the drug carrier and the nanoparticle’s core [119]. In a different study, Mehta created ultrasonic-sensitive nanoparticles by encasing hydrophobic dyes such as ICG and Nile Red in HP-β-CD cavities and then using poly(benzyl methacrylate) as the support structure. These gas-cavity nanoparticles were created for AS treatment and diagnosis [13].

### 5.3. Lipid Nanoparticles

Liposomes are small, spherical vesicles composed of phospholipid bilayers capable of encapsulating hydrophilic and lipophilic substances [120]. CD can be integrated as a functional unit into functionalized liposomal structures to form nanoparticle delivery systems for AS therapy. In this system, CD can adopt two integration modes: one incorporated within the liposome’s lipid bilayer and the other by being anchored to the surface of the liposomal bilayer via host–guest recognition. This enables the system to acquire new functionalities, such as stimulus-response and targeting capabilities. In nanoparticle systems aimed at targeted treatment for AS, CD is typically introduced as a functional unit to the liposome surface, leveraging dynamic host–guest recognition mechanisms to facilitate stimulus responses and controlled release.

In treating AS, reconstituted high-density lipoprotein is suitable for constructing liposomes. For instance, Liu et al. developed a system that used TPGS-modified β-CD and PEGylated ferrocene to recognize the host and form a supramolecular polymer known as PF/TC. When this polymer was introduced into the lipid bilayer of rHDL, CD was a crucial surface functional unit that allowed the system to react to reactive oxygen species (ROS) [121]. Modifying the surface of liposomes with CD to construct functionalized liposomes for effective drug delivery and functional enhancement. Key components included 1,2-distearoyl-sn-glycero-3-phosphoethanolamine-polyethylene glycol-adamantane, providing specific biocompatibility and targeting properties to the liposomes. β-CD, as the host molecule, can bind with adamantane on the liposome surface and dissociate in pathological environments, exposing active units [122]. Similarly, in a series of studies by He et al., reconstituted high-density lipoprotein (rHDL) nanoparticles were constructed [123], where CD was covalently linked with TPGS-β-CD to form copolymers inserted into the liposome surface, creating β-CD-anchored carriers that provide further modification sites for the system [124].

### 5.4. Nanogels

Nanogels are tiny gel-like particles with nanoscale dimensions commonly used as drug delivery carriers [125]. Their advantages include increasing drug solubility, stability, and bioavailability, enabling targeted delivery and controlled release, enhancing therapeutic efficacy, and reducing side effects [126]. By binding CD-NH_2_ to poly(2-(N-morpholino)ethyl methacrylate), a ROS-sensitive polymer, and then loading it with the anti-inflammatory medication prednisone and a fluorescent probe, Xu et al. produced a nanogel [75], loading anti-inflammatory drug prednisone and AIE probe lithium iron phosphate into the hydrophobic cavity. Subsequently, aldehyde-modified dextran was crosslinked with CD-NH_2_ to form nanogels for AS treatment. In this system, the AIE probe exhibited a blue shift in emission wavelength within the lipid environment, achieving precise localization of atherosclerotic plaques. The poly(2-(N-morpholino)ethyl methacrylate) polymer transitioned from hydrophobic to hydrophilic in ROS environments, triggering nanoparticle disassembly and releasing the anti-inflammatory drug prednisone. CD acted to clear lipids from the plaques through host–guest recognition, forming a dual-action therapeutic strategy with prednisone [127]. Similarly, Shoukat et al. created interpenetrating polymer network nanogels by combining synthetic polyvinyl alcohol (PVA) with natural β-CD. Compared to conventional forms, these were produced by radical polymerization, which significantly increased the solubility and release of rosuvastatin [128].

### 5.5. Inorganic/Metal Hybrid CD Systems

Inorganic/metal hybrid CD systems and CD-based metal–organic frameworks (CD-MOFs) integrate the chemical properties of metals with CDs’ molecular recognition capabilities, exhibiting shared characteristics like high porosity, large surface area, and biomolecule affinity. These emerge from CDs’ role as organic ligands, utilizing their abundant hydroxyl groups for coordination with metal ions or inorganic components [129]. Specifically, γ-CD, a C8 symmetric cyclic oligosaccharide with a hydrophobic cavity (~1 nm diameter, ~0.8 nm depth), leverages its external hydrophilicity and internal hydrophobicity to form coordination bonds with ions such as K^+^, constructing porous frameworks via self-assembly. This coordination underpins MOF formation, granting structural diversity and functional tunability, which are critical for targeted therapies [130].

CD-MOFs address pathological challenges in AS through three key mechanisms: cholesterol crystal dissolution via a “molecular sponge” effect that reduces plaque deposition; anti-inflammatory and antioxidant actions by delivering drugs to lesions while synergizing with CDs’ innate activities to inhibit cellular stress; and enhanced drug delivery by loading statins or gene therapies into tunable pores for sustained, targeted release, improving bioavailability and reducing toxicity. Supporting studies include Shi et al.’s work where CM-β-CD coordinated with Fe_3_O_4_ nanoparticles to stabilize IL-10 for MRI-tracked delivery, and Kang et al.’s β-CD functionalized mesoporous silica enabling stimulus-responsive release through host–guest complexes. These approaches demonstrate CD-MOFs’ potential in precise AS treatment by combining structural advantages with therapeutic efficacy [131,132].

In conclusion, CD is a valuable and adaptable component of CD-NDDS (Figure 3). Through self-assembly or nano-precipitation, it can serve as a crucial component in the formation of nanoparticles. Because of its host–guest recognition capabilities, it can also give nanoparticle systems new functions, such as targeted delivery, slow release, or response to internal or external stimuli. The following chapters will provide a detailed explanation of how CD enhances these systems’ therapeutic qualities.

## 6. Design of CD-NDDS Targeting for AS

In targeted nanodelivery systems for AS, CD-NDDS can be further functionally designed to impart targeting capabilities beyond their fundamental structures, such as micelles, polymeric nanoparticles, liposomal nanoparticles, and nanogels. The most widely used design strategy involves functional modifications of the surface structures of the nanocarriers, typically employing surface modification strategies such as chemical modifications [133], physical modifications [134], and biomimetic adaptations [135].

### 6.1. RGD Modification

To address the specific acidic (low pH [136]) and reactive oxygen species (ROS [136]) Microenvironment characteristic of vascular inflammation, Zhang et al. designed a dual-responsive nanocarrier that responds to both pH and ROS. Usingβ-CD as the backbone, they constructed two types of CD-functionalized materials: ACD and OCD, which were integrated into PLGA nanoparticles via a nanoprecipitation method. On this basis, they utilized the self-assembly features of amphiphilic phospholipids to modify phospholipid components on the nanoparticle surface through hydrophobic interactions, electrostatic interactions, and entropy-driven forces, thereby providing structural stability to the nanocarrier, extending blood circulation time, and anchoring the RGD targeting ligands to the material surface [136]. Similarly, Kang et al. created adamantane-PEG-RGD caps by utilizing RGD peptides as targeting elements and affixing them to adamantane via PEG chains. To create the intended delivery system, these caps were placed inside the β-CD cavity [137].

### 6.2. HA Modification

Hyaluronic acid (HA) is a widely used essential natural material for constructing targeted inflammatory microenvironments, possessing a specific binding ability to overexpressed CD44 receptors [138]. In the nanoparticles designed by Ma et al. Polymeric Light-Coupled and Dual-responsive Probe incorporated in Polymersome Micelles (Polymeric Nanoplatform), β-CD loads the glucocorticoid prednisolone through host–guest recognition to form an L-Cysteine-desulfhydrase complex embedded within the nanoparticle interior, while the surface of the nanoparticles is coated with oxidized hyaluronic acid (oxHA) for specific binding to the CD44 receptors overexpressed at the plaque sites [139]. Similarly, He et al. attached a phosphatidylserine-targeting peptide (PTP) to CD molecules to create self-assembled micelles. To target both CD44 receptors and apoptotic cells, HA was electrostatically coated onto the micelles’ surface [140]. Yu et al. conducted similar research by linking CD to HA and obtaining CD nanoparticles with inflammation-targeting properties through an emulsification-solvent evaporation method [113]. In another study by He et al., CDs were added to the liposome surface, while HA-Fc was grafted onto the nanoparticle via host–guest recognition to allow for effective targeting [123].

### 6.3. Cell Membrane Modification

A biomimetic modification strategy utilizing cell membrane coverage is also a viable approach to address the inflammatory microenvironment in AS [141]. Zhu et al. created an amphiphilic block polymer that self-assembled into core–shell micelles by attaching dopamine and β-CD to the polymer. CD served as the hydrophobic core, and dopamine as the surface in this structure. Dopamine’s positive charge enabled it to adsorb macrophage membranes electrostatically, coating the nanoparticle and giving it the ability to target inflammation [142]. In another paper, Gao et al. used the host–guest recognition between β-CD and adamantane to couple liposomes loaded with quercetin to macrophages for targeted therapy against AS [143]. Similarly, Zhu et al. utilized the hydrophobic cavity of MPEG-β-CD to interact with the lipid tail of MM, successfully encapsulating the macrophage membrane on the surface of MPEG-β-CD micelles (Figure 4) [102]. This biomimetic system exploits the inflammation chemotactic effect of macrophage CCR2/CD47 to target inflammation at the thrombus site [102,144]. Additionally, Shi et al. synthesized carboxymethyl-β-CD-modified magnetite nanoparticles as carriers for IL-10, covering their surface with macrophage membranes through electrostatic and hydrophobic interactions [145].

However, to facilitate their clinical application, several challenges must be addressed for biomimetic-CD NDDS, including issues related to poor reproducibility in fabrication, inadequate membrane stability, and the complexities of immune clearance mechanisms in diverse in vivo environments. Future research should concentrate on developing standardized techniques for cell membrane extraction and nanoparticle coating, improving the long-term stability of these membranes, and deepening our understanding of the interactions between cell membrane biomimetic nanoparticles (CMBNs) and the host immune system. This approach will aid in designing safer and more effective biomimetic CD-NDDS.

## 7. Applications of CD-NDDS in the Therapy of AS

CD demonstrates a distinctive ability to act as both “a drug carrier and therapeutic agent” in nanodelivery systems used for AS treatment. Its hydrophobic cavity effectively encapsulates anti-atherosclerotic drugs such as statins [146] and anti-inflammatory agents [147], which helps in enhancing their solubility and stability. Furthermore, CD exerts direct therapeutic effects by dissolving CCs, activating liver X receptors (LXR) to mediate cholesterol efflux, and downregulating the expression of inflammatory factors [148]. Furthermore, CD derivatives, like sulfobutyl-β-CD, can combine with lipids or polymers to create specific nanoparticles, such as membrane-coated systems that mimic biological processes [149]. When exposed to acidic or high ROS environments, these nanoparticles can release their contents and accumulate in plaques using the Enhanced Permeability and Retention (EPR) effect. This makes it possible to simultaneously have anti-inflammatory, cholesterol-lowering, and plaque-stabilizing impact [150]. This dual function makes CD both a carrier and an active therapeutic agent (Figure 5). Such a “drug release + microenvironment remodeling” strategy provides innovative insights into AS treatment [151]. In the following sections, this review will examine the molecular mechanisms of CD in this context, focusing on its formulation and pharmacological behavior in treating AS (as seen in Table 1).

### 7.1. Formulation Properties of CD Within the NDDS

As a key part of nanodelivery systems, CD helps in improving the solubility of hydrophobic drugs through host–guest recognition, effectively overcoming prevailing drug delivery challenges [157]. CD also optimizes drug release kinetics by utilizing steric hindrance and mechanical strength to reduce drug diffusion rates [117]. Also, chemical changes enable CD to react to environmental stimuli, creating controlled-release cores that release medications in response to variations in light, pH, or reactive oxygen species (ROS), enabling targeted treatment of diseased regions [158]. This shift from static solubilization to dynamic responsiveness represents a modern and promising direction in treating AS.

#### 7.1.1. Solubilization

The distinctive ring-shaped structure of CD enables it to use hydrophobic forces to trap poorly soluble drug molecules and transform them into water-soluble complexes, which significantly improves drug solubility. Notably, when combined with nanoscale carriers, substantial improvements in drug loading capacity can be achieved, enabling targeted delivery and sustained release to atherosclerotic plaques. For instance, the host–guest recognition between α-CD and poly(ethylene glycol)-polylactic acid micelles facilitate the construction of a supramolecular hydrogel system. Experimental evidence indicates that the inclusion complex crystals formed between α-CD and PEG chains significantly enhance drug loading capacity [156]. By creating an inclusion complex of Cantharidin with HP-β-CD, this system achieved a remarkable increase in drug solubility [159].

Many medications (such as methotrexate or statins) are poorly soluble because of their high hydrophobicity, as AS is linked to problems with fat metabolism and chronic inflammation. Particularly for statins or paclitaxel, CD can stop these medications from crystallizing and enhance their chemical stability, significantly improving how well the body absorbs them [160]. The CD nanodelivery system integrates solubilization strategies through host–guest encapsulation and micellar embedding, with the nanoscale structure potentially increasing drug solubility by 10 to 100 times. In these systems, the hydrophobic cavity of CD encapsulates diverse hydrophobic drug molecules (e.g., methotrexate, atorvastatin calcium, prednisone, and phenylbutyric acid) or diagnostic therapeutic agents (such as Luminol, imidazole derivatives), forming interactive complexes. During subsequent processing, CD-drug inclusion complexes are integrated as functional units into micelles, liposomes, and polymeric nanoparticles, thus further enhancing drug stability.

#### 7.1.2. Controlled Release

The unique molecular structure of CD establishes a foundation for its controlled release functionality. CD forms “lock-and-key” inclusions with drug molecules by geometric compatibility. This property is particularly prominent in cholesterol transport, where the β-CD cavity aligns perfectly with the cholesterol steroid nucleus, permitting stable release of the complex in physiological conditions [161]. Additionally, inorganic composites, such as silica nanoparticles loaded with α-CD, can facilitate demand-based release through photothermal effects [162].

In developing nanocarriers, CD enhances controlled release functionalities through structural innovations. The density of the cross-linked network directly influences the capacity for drug retention. For instance, high cross-link density nanosponges create steric hindrance to control drug release. Hybrid polymer approaches, such as CD polymers, utilize covalent cross-linking to form three-dimensional networks (like the β-CD/benzoate system), enhancing structural stability and enabling prolonged sustained release [163].

In AS treatment, β-CD forms inclusion complexes with rosuvastatin through its hydrophobic cavity. Additionally, polystyrene-modified β-CD self-assembles into physical cross-linking points via hydrophobic interactions, producing a molecular size-selective barrier within β-CD to restrict unregulated drug diffusion and promote controlled release [128]. Xu et al. constructed a multifunctional gel for targeted diagnostic and therapeutic applications in AS [164]. In this formulation, CD interacts with the hydrophobic chain of poly(2-(N-morpholino)ethyl methacrylate) via host–guest recognition, yielding a dynamic physical cross-linked network at the nanoparticle core. Concurrently, the amine groups of CD-NH_2_ undergo a Schiff base reaction with the aldehyde groups of ox-Dex, generating a covalent cross-linked network that enhances nanoparticle rigidity and regulates drug release through dynamic dual cross-linking [164].

Moreover, it is notable that CD nanodelivery systems display affinity-driven controlled release characteristics. In a study conducted by Kim et al., a system was created to release medications in environments high in cholesterol by taking advantage of the fact that cholesterol binds to CD more strongly than simvastatin does [165]. In this mechanism, CD assumes a tripartite role: acting as a drug carrier, cholesterol capture agent, and structural component of nanoparticles. Once coding single nucleotide polymorphism (CSNP) enters a cholesterol-enriched plaque microenvironment, the binding constant of cholesterol to CD is significantly greater than that of simvastatin. The cholesterol displaces the drug when these nanoparticles enter cholesterol-filled plaques because the cholesterol binds to CD more firmly, causing the drug to be released while simultaneously eliminating excess cholesterol (see Figure 6) [165].

#### 7.1.3. Stimulus Response

The dynamic recognition mechanism inherent in host–guest recognition allows CD-NDDS to respond to external stimuli. First, functioning as a molecular “switch”, its hydrophobic cavity directly regulates drug release kinetics through host–guest encapsulation, notably enhancing drug expulsion rates [166]. Secondly, chemical modifications (such as sulfonic and carboxylic groups) confer environmental sensitivity on CD, transforming it into a responsive trigger [167]. For instance, in acidic environments, carboxymethylated CD gets protonated, increasing the release of Chol and accelerating drug dissociation [103].

It is common practice to use high ROS levels in inflammatory areas when designing drug delivery systems that react to natural triggers. For instance, in the PF/TC-AT-d-rHDL nanoplatform, β-CD forms ROS-sensitive host–guest complexes (Fc@β-CD) by binding to hydrophobic ferrocene (Fc). When there is low ROS in the bloodstream, this structure remains stable; however, when there is high ROS in plaque areas, the complex disintegrates and the drug is released in response to disease signals. Likewise, the poly-β-CD/pH-sensitive benzimidazole-modified dextran sulfate/spherical nucleic acid system, which is composed of polyβ-CD grafted with benzimidazole and assembled with polysaccharide sulfate, uses the CD cavity as a “switch” that remains intact in healthy conditions but disintegrates in diseased ones [123]. In addition, in ROS-responsive nanostructures (HA-Fc/NP_3_ST), β-CD facilitates reversible size alterations in nanoparticles through host–guest recognition, acting as a key mechanism that triggers disintegration upon ROS exposure, consequently releasing smaller particles for deeper tissue penetration (Figure 7) [123]. Research has demonstrated the construction of dual-responsive nanoparticles to pH and reactive oxygen species (ROS) by combining pH-sensitive materials (ACD) with oxidation-responsive materials (OCD), as shown in Figure 6e. OCD can be readily synthesized through the chemical functionalization of β-CD. By simply adjusting the weight ratio of ACD and OCD, the pH/ROS responsiveness can be easily tuned, yielding nanoparticles with distinct hydrolysis characteristics under inflammatory microenvironments. As shown in Figure 7f–j, ACD nanoparticles exhibited significantly accelerated hydrolysis at both pH 5 and pH 6, with negligible influence from hydrogen peroxide presence. Regardless of pH, hydrogen peroxide significantly accelerated the hydrolysis of OCD nanoparticles. Conversely, OCD nanoparticles exhibited similar hydrolysis curves at pH 5, 6, or 7.4. When tested in different buffers, the hydrolysis behavior of nanoparticles derived from both ACD and OCD was influenced by both pH and H_2_O_2_. These data reveal the dual pH/ROS responsiveness of ACD/OCD-based nanoparticles. In summary, a pH-ROS dual-responsive nanoplatform was successfully constructed by simply combining the pH-responsive material ACD with the ROS-unstable material OCD. Notably, the precise response sensitivity of the nanocarrier can be easily tuned by altering the mass ratio of ACD/OCD, endowing it with robust scalability—an additional advantage for translational applications [136].

### 7.2. Pharmacological Properties of CD in the NDDS

CD has a special hydrophobic cavity structure that allows for efficient cholesterol removal. This structural characteristic enables it to selectively encapsulate Chol sterols through geometric fitting and non-covalent interactions, such as hydrogen bonds and van der Waals forces, effectively extracting free Chol and crystals from cell membranes or lysosomes. For instance, HP-β-CD can disrupt the lattice structure of CCs, significantly enhancing their solubility [168]. Therefore, CD’s pharmacological function in nanodelivery systems primarily consists of removing cholesterol from target sites, maintaining balanced cholesterol levels, dissolving CCs, promoting cholesterol outflow, and lowering inflammation—all of which aid in treating AS [124].

#### 7.2.1. Dissolving CCs

CD-NDDS exhibits significant advantages in dissolving CCs within atherosclerotic plaques due to the unique cavity structure of CD, which endows superior molecular recognition capabilities, allowing for targeted binding to Chol molecules [13]. Additionally, chemical modifications can significantly enhance the functionality of these systems, thereby improving their ability to eliminate Chol selectively [169].

For instance, β-CD nanosponges leverage a large-scale three-dimensional porous structure formed through cross-linking, significantly enhancing CD’s ability to adsorb free Chol and small CCs. Their dissolution efficiency is much higher than that of monomeric CD, while also considerably reducing cytotoxicity [170]. Similarly, the biomimetic nanoplatform HT-rHDL effectively exploits the natural targeting ability of HDL to deliver CD precisely to plaque cores. In this context, the CD cavity reduces solution supersaturation through multiple binding sites, which enhances the disintegration efficiency of CCs [169].

This multi-mechanism collaborative strategy presents a novel nanotechnology approach for AS treatment. Responsive structures, like the poly-β-CD/pH-sensitive benzimidazole-modified dextran sulfate/spherical nucleic acid supramolecular assemblies, utilize the directional encapsulation capacity of CD, allowing for targeted wrapping of crystal surfaces under specific conditions. Consequently, the CD cavity plays a critical role in disrupting the hydrogen bonding network among CCs, enhancing water molecule permeation and ultimately leading to a substantial reduction in the volume of CCs within plaques [103]. The Chol recognition capability of CD is further amplified through surface multivalent arrangements; for example, magnetic nanoparticles modified with β-CD (Fe3O4@CM-β-CD) can bind Chol at capacities of up to 291 mg/g of polymer material at a specific concentration, the polymer material can bind cholesterol at a high capacity, which is much higher than that of the free β-CD system [171].

Notably, research shows that there is a novel biosimulated targeted delivery system based on CD, which adopts a free-rider strategy [172]. They incorporated DSPE-PEG-β-CD into macrophage membranes for surface modification. In this context, β-CD is a drug carrier and a bioactive molecule directly involved in CC clearance, promoting Chol efflux via physical encapsulation and biological regulation. In vitro experiments demonstrated that macrophages treated with DSPE-PEG-β-CD exhibited significantly reduced CC levels [143]. In in vivo models, the functionality of β-CD was enhanced through nanocarrier design, effectively transforming macrophages into “mobile clearing units” that target atherosclerotic plaques via inflammatory chemotactic effects. This approach markedly enhanced the clearance rates of CCs, which was also corroborated in clinical samples. Human carotid plaques treated with DSPE-PEG-β-CD combined with QT-NP displayed a reduction in Chol content (Figure 8) [143].

#### 7.2.2. Promoting Cholesterol Efflux

CD-NDDS greatly improves the ability of macrophages to promote Chol efflux by integrating CD’s unique molecular recognition capabilities with advanced carrier engineering technologies. This offers new pathological intervention strategies for AS [102]. Significantly, the physicochemical properties of CD directly influence its biological effects. At low concentrations of β-CD, its superior Chol binding-dissociation kinetics render it particularly suitable for constructing dynamic transport systems [173]. Nevertheless, CD may trap Chol inside at higher concentrations, decreasing its transport efficiency.

Biomimetic nanoparticles, such as macrophage membrane-coated poly-β-CD nanoparticles (MM@MTX NPs), target atherosclerotic plaques through SR-A1 receptor [174]. In addition to dissolving (CCs), the inner β-CD increases ABCA1/SR-B1 expression in conjunction with methotrexate [175], facilitating Chol’s exit from the cell via SR-BI pathways [176]. This demonstrates how co-loaded medications and CD can promote cholesterol transport.

Additionally, CD controls gene pathways that aid in the removal of cholesterol. For instance, DSPE-PEG-β-CD increases ABCA1 and ABCG1 levels by activating the LXR pathway, which improves macrophage cholesterol efflux. The biological function of CD is directly linked to this effect [100]. In a different example, Zhu et al. created a biomimetic CD-NDDS [177] (MM@DA-pCD@MTX) wherein methotrexate activates important genes like CYP27A1 and ABCA1 to promote Chol transporter proteins, while CD captures CCs. Chol can exit the plaques more effectively thanks to the dual action of the drug’s gene activation and CD’s physical removal. The macrophage-coated NPs increased ABCA1/CYP27A1 expression surrounding plaques in animal models, accelerating reverse transport of Chol and enhancing transporter activity (Figure 9) [177].

Another tactic, the “shuttle-esterification” dual-functional model, employs β-CD-functionalized simvastatin loaded onto reconstituted high-density lipoprotein (β-CD-ST-d-rHDL), demonstrating high drug delivery and Chol efflux in vitro [178]. In this case, CD functions as a helper molecule, removing free Chol from the membrane and generating a gradient that facilitates its exit from the cell. Concurrently, LCAT enzymes on rHDL convert free cholesterol into esters, facilitating reverse transport.

In another study, mannose-modified β-CD (MAN-βCD) was used to target mannose receptors (CD206) on macrophage surfaces. This compound disrupts lipid raft structures and activates the ABCA1/ABCG1 Chol efflux pathway, ultimately enhancing Chol efflux efficiency compared to baseline levels [52]. Similarly, host–guest interactions create β-CD/adamantane-based liposomes that target macrophages, enabling the CD to retain Chol and trigger the LXRα/ABCA1 pathway for efficient transport [143].

#### 7.2.3. Alleviating Inflammatory Microenvironments

CD-NDDS modifies the inflammatory microenvironment of AS by removing deleterious factors, altering immune cell behavior, and collaboratively inhibiting related signaling pathways, leveraging the unique molecular recognition and regulatory capabilities of CD. The underlying mechanism chiefly involves the regulation of macrophage polarization [179].

CD-NDDS directly regulates macrophage polarization by activating intrinsic cytoprotective mechanisms. The oxidative-responsive β-CD derivatives (Ox-bCD NPs) significantly enhance Nrf2 nuclear translocation and upregulate downstream antioxidant genes NQO1 and HO-1 under H_2_O_2_ stimulation [180,181,182]. This polarization mechanism originates from CD’s biological activity and promotes the transition to anti-inflammatory M2 phenotype through well-defined biochemical pathways, demonstrating CD’s capacity to directly reprogram macrophage function in atherosclerotic environments.

CD-NDDS facilitates M2 polarization by remodeling the inflammatory microenvironment through multiple approaches. Supramolecular systems utilizing CD’s host–guest recognition, such as polydopamine nanoparticles complexed with adamantane-modified β-CD, effectively scavenge reactive oxygen species (ROS) in macrophage models [183,184]. This oxidative stress reduction significantly decreases pro-inflammatory cytokine secretion (TNF-α and IL-6) induced by LPS, establishing a less inflammatory microenvironment that supports M2 polarization [185,186]. Additionally, the unique molecular structure of CD enables targeted drug release in damaged tissues, inhibiting ox-LDL oxidation and preventing foam cell formation through ROS reduction in macrophages [187].

Dual-responsive carriers (ACD/OCD systems) leverage CD’s ability to load hydrophobic drugs and release them specifically in acidic, high-ROS inflammatory vasculature [136]. These systems respond to vascular inflammatory microenvironments, inhibiting PDGF-BB-induced smooth muscle cell migration and proliferation through downregulation of pro-inflammatory cyclin D1 [188]. In response to vascular injury, the migration of vascular smooth muscle cells (VSMCs) from the media to the intima plays a crucial role in the development of restenosis, while the Dual-responsive-CD NDDS significantly inhibits vascular smooth muscle cells migration [189,190]. Furthermore, the carriers achieve selective accumulation at lesion sites, significantly reducing oxidative damage markers (8-OHdG) and degrading enzymes (MMP-2) while neutralizing ROS and restoring antioxidant enzyme activity (Figure 10).

## 8. Challenges and Perspectives

Despite the vast potential of CD-NDDS as therapeutic agents against AS, numerous hurdles still prevent their practical application in a clinical context: these primarily concern biocompatibility, delivery specificity, pharmacokinetics, and mass production.

CD derivatives, while promising for therapeutic applications, present significant safety challenges that necessitate careful management. HP-β-CD demonstrates ototoxicity at high doses, which damages inner ear hair cells and leads to hearing loss. Additionally, its renal excretion pathway imposes a burden on the kidneys with prolonged use, heightening the risk of nephrotoxicity. Chronic administration of HP-β-CD results in the formation of crystalline deposits in renal tubules [191]. Moreover, the non-selective cholesterol extraction capability of CD’s hydrophobic cavity can induce hemolysis and modify erythrocyte lipid structures, as evidenced by M-β-CD. Current encapsulation and polyamidation technologies can prolong circulation time but introduce immunogenic risks with repeated exposure. To address these challenges, Sahar Roozbehi and colleagues employed enzyme-responsive poly-CD designs to reduce toxicity [192,193]. On the other hand, the incorporation of anionic groups into molecules such as SBE-β-CD [194], can also help to reduce toxicity to some extent.

The regulatory approval of CD derivatives by the food and drug administration (FDA) and european medicines agency (EMA) hinges on rigorous safety profiling, intended formulation use, and thorough toxicity assessments. However, clinical translation faces significant hurdles in manufacturing consistency and preclinical model relevance. Synthetic challenges, such as multi-step derivatization and imprecise substitution control (e.g., acetylation instability affecting drug loading uniformity [195]), directly impact product quality and regulatory compliance. Cross-linked CD polymers exhibit heterogeneous molecular weight distributions, leading to inconsistent cholesterol-binding site stability and variable in vivo efficacy [7,196]. Preclinical models like ApoE KO mice show limited translational fidelity due to disparities in fibrous cap thickness and calcification compared to human plaques [197]. To address these gaps, microfluidic systems enable scalable production of uniform nanoparticles via controlled flow rates [198], while non-human primate models (e.g., baboons with hyperlipidemia-induced lesions [199]) better mimic human pathophysiology, offering more reliable platforms for regulatory endorsement. Ultimately, aligning manufacturing precision and model selection with regulatory standards is crucial for advancing CD-based therapies.

The advancement of new delivery systems is crucial for overcoming existing limitations. Current responsive designs frequently fail in the heterogeneous microenvironments of plaques. For example, the pH-sensitive poly-β-CD/pH-sensitive benzimidazole-modified dextran sulfateid is effective under lysosomal acidic conditions, yet fluctuations in local pH may prematurely terminate drug release [103]. In addition, dense fibrous caps and calcified regions obstruct nanoparticle penetration, such as macrophage membrane-coated MM-pCD NPs. At the same time, passive targeting can lead to nonspecific accumulation in the liver and spleen [200,201,202]. Future strategies should incorporate innovative mechanisms, including developing pH/ROS dual-responsive systems to address a broader spectrum of pathological microenvironments [203]. Furthermore, integrating diagnostic and therapeutic designs may facilitate simultaneous treatment and real-time efficacy assessment, providing dynamic feedback for precise intervention.

Aside from AS, lipid metabolism targeting creates new opportunities as cancer cells rely on cholesterol-rich areas (rafts) to support survival signals (such as the EGFR/PI3K pathways), cholesterol imbalance causes AS, and is associated with tumor growth [204]. CD can selectively extract Chol from membranes, compromising raft integrity and inhibiting tumor cell proliferation and metastasis [205]. Furthermore, delivery methods that resemble high-density lipoprotein (rHDL) have already demonstrated efficacy in eliminating cholesterol from AS foam cells and may be helpful in cancer treatment. For instance, adding LXR agonists to these systems can improve the removal of cholesterol from tumor-associated macrophages (TAMs), activate ABCA1, and transform them from an immunosuppressive to an immune-supporting state [206]. This shared regulatory pathway between AS and tumors creates a new direction: using CD-NDDS to develop “metabolism + immune” dual-target therapies against cancer.

Artificial Intelligence (AI) is rapidly advancing CD molecule design. Machine learning algorithms predict CD-guest binding affinity, optimizing drug delivery systems. QSAR models integrated with ML support functional prediction [207,208]. AI analyzes protein pores to design specific-binding CD analogs. Future generative models will design derivatives from scratch, expanding molecular space [209]. Combined with high-throughput screening, AI enables automated design for drug delivery and pollutant removal. In drug development, AI shortens R&D by predicting ADMET properties [210]. With improved algorithms and compute, AI will transform CD applications in medicine and nanotechnology, enabling efficient custom high-performance CDs.”

## 9. Conclusions

This study systematically reviews the strategies associated with the CD nanodelivery system for targeting Chol removal in treating AS, emphasizing the exceptional adsorption properties of CD. Moreover, it outlines the future development directions. The CD-NDDS employs an innovative “drug-excipients-in-one” design concept, offering effective intervention measures for addressing the pathological cascades arising from Chol metabolism disorders, crystal deposits, and inflammatory activation in AS. The system enables the targeted breakdown and biochemical transformation of cholesterol CCs by leveraging molecular interactions between CD and Chol. It also facilitates intelligent nano-engineering to modify the inflammatory microenvironment. The system allows for the precise dissolution and metabolic remodeling of CCs by utilizing the molecular interactions between CD and Chol. It also facilitates intelligent nano-engineering to modify the inflammatory microenvironment.

Nevertheless, significant bottlenecks persist in enhancing biocompatibility, optimizing targeted delivery efficiency, and advancing clinical translation. Therefore, future research should focus on exploring interdisciplinary integration strategies to develop smart carriers with both diagnostic and therapeutic functions, establishing continuous manufacturing processes based on microfluidic technology. Leveraging the advantages of microfluidics: First, precise control and efficient synthesis: Microfluidics enables precise regulation of nanoparticle size, morphology, and functionality through micro-scale fluid management, achieving uniform, scalable production while reducing batch-to-batch variability and purification requirements. Second, multifunctional carrier development: supporting integrated manufacturing of stimulus-responsive particles, such as microfluidic chip-based IL-10 delivery systems. Continuous process optimization enhances delivery kinetics (release rate, plaque penetration) to improve therapeutic precision; Third, driving therapeutic paradigm shifts: transitioning from traditional “cholesterol clearance” to “restoring microenvironment homeostasis.” Microfluidics serves as the core enabling tool to accelerate clinical translation of smart carriers. These advantages are poised to propel the technology from a simple “cholesterol clearance” model toward a revolutionary therapeutic paradigm centered on “restoring vascular microenvironment homeostasis”.

## Figures and Tables

**Figure 1 pharmaceutics-17-01496-f001:**
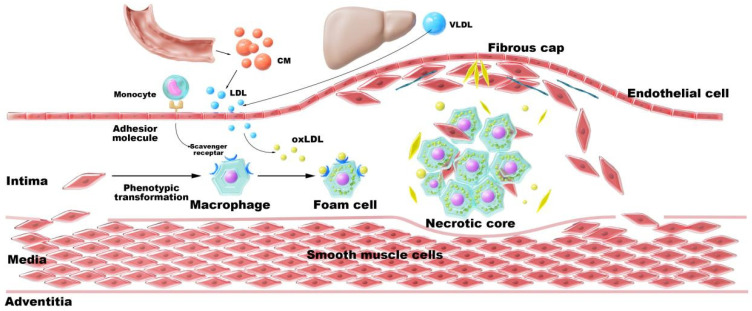
Pathological cascade of cholesterol dysregulation in atherosclerosis. The figure was originally created by the authors using Adobe Photoshop 2020.

**Figure 2 pharmaceutics-17-01496-f002:**
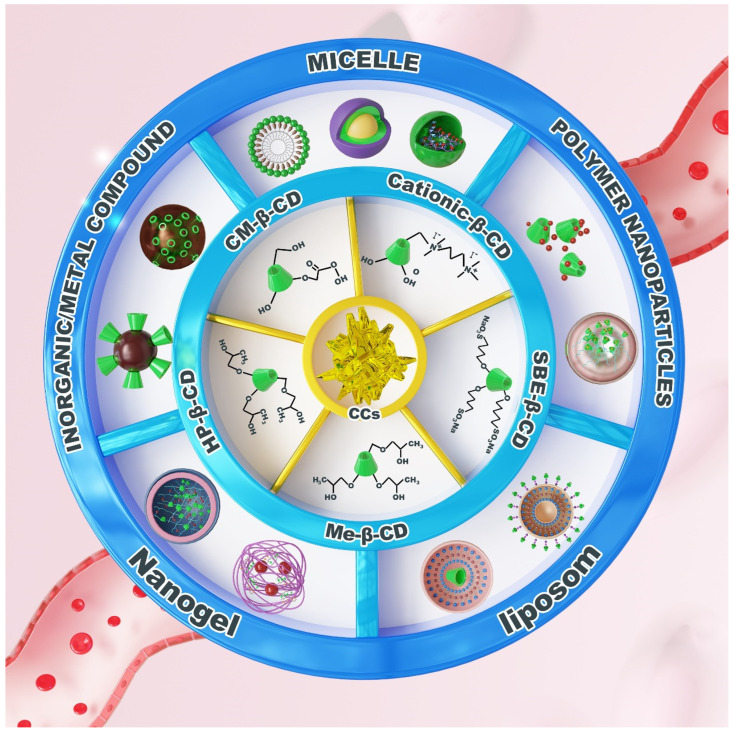
Diversity of CD-engineered nanocarriers for AS therapy. Micelles formed via host–guest recognition between CD-conjugates and hydrophobic guests (e.g., cholesterol), enabling drug solubilization and plaque targeting; Polymeric nanoparticles integrating CD through covalent or host–guest interactions with polymers like poly(lactic-co-glycolic acid) (PLGA), offering multifunctional drug loading and controlled release; Lipid nanoparticles featuring CD-embedded bilayers or surface modifications for stimuli-responsive targeting (e.g., ROS sensitivity); Nanogels composed of CD-crosslinked networks (e.g., dextran or poly(2-(N-morpholino)ethyl methacrylate)) that respond to pathological signals (e.g., ROS) for plaque-specific drug release; and Inorganic/metal hybrids combining CD with materials such as Fe_3_O_4_ or mesoporous silica for theranostic applications (e.g., MRI and drug delivery). These nanocarriers leverage CD’s dual role as a structural scaffold and functional modulator to synergistically dissolve cholesterol crystals, promote cholesterol efflux, and alleviate inflammatory microenvironments in atherosclerotic plaques. The figure was originally created by the authors using Adobe Photoshop 2020.

**Figure 3 pharmaceutics-17-01496-f003:**
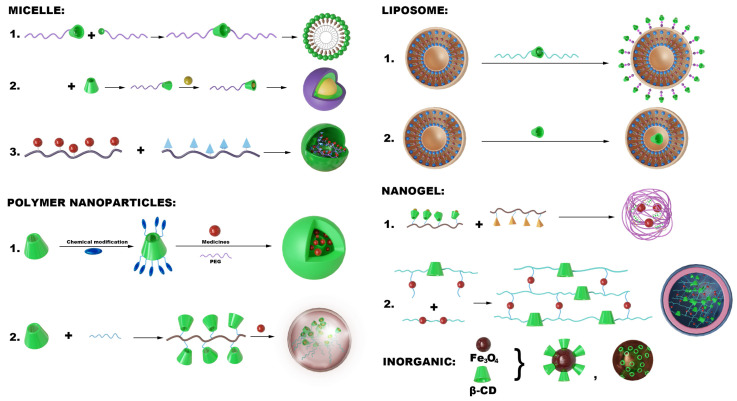
CD-engineered nanocarriers enable targeted AS therapy through: (1) For micelles—CD conjugates forming drug-loaded micelles via host–guest recognition; (2) For polymeric NPs—Host–guest/covalent CD-polymer integration for multifunctional PLGA systems; (3) For lipid NPs—CD-embedded bilayers/surface anchors conferring stimulus-responsive targeting; (4) For nanogels—ROS-responsive CD-monomer networks achieving plaque-specific release; (5) For inorganic hybrids—CD-metal complexes (e.g., CM-β-CD/Fe_3_O_4_) combining drug delivery with MRI. CD serves dual roles as a structural scaffold and functional modulator. The figure was originally created by the authors using Adobe Photoshop 2020.

**Figure 4 pharmaceutics-17-01496-f004:**
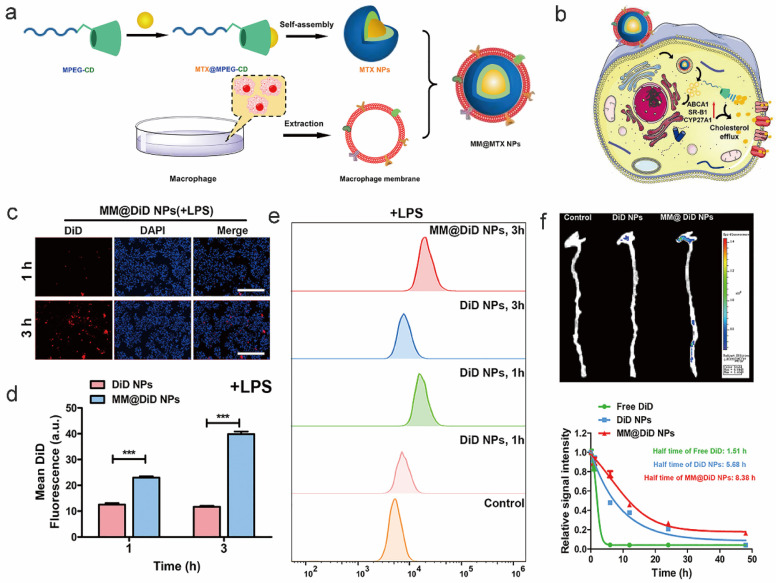
Illustrations of MM@MTX NPs targeting thrombotic inflammatory sites through macrophage membrane modification [102]. (**a**): The preparation schematic of MM@MTX NPs; (**b**): The MM@MTX NPs target inflammatory macrophages via surface CCR2/CD47; (**c**,**d**): Fluorescence microscopy results demonstrate that MM@MTX NPs have enhanced uptake efficiency in inflammatory macrophages; (**e**): Flow cytometry analysis reveals that MM@MTX NPs exhibit superior uptake efficiency in inflammatory macrophages; (**f**): Compared with DiD NPs group having a strong fluorescence in the liver and little fluorescence in the lung and aorta, MM@DiD NPs could significantly reduce the undesirable delivery and accumulation into the liver. This indicates that MM@MTX nanoparticles significantly enriched at thrombotic sites and exhibited higher plasma concentrations. DiD: A lipophilic fluorescent dye commonly used to label cell membranes; here it is employed to label nanoparticles (NPs) for observing their intracellular distribution. DAPI: 4′,6′-Diamidino-2-phenylindole, a fluorescent dye that binds to DNA; Merge: Indicates the superimposition of images from the DiD and DAPI channels, allowing simultaneous visualization of nanoparticle distribution (DiD signal) and nuclear localization (DAPI signal); Statistical Significance Markers: In scientific research, *** indicates *p* < 0.001. These denote statistically significant differences between groups, with more asterisks indicating greater significance. Reprinted from Ref. [102].

**Figure 5 pharmaceutics-17-01496-f005:**
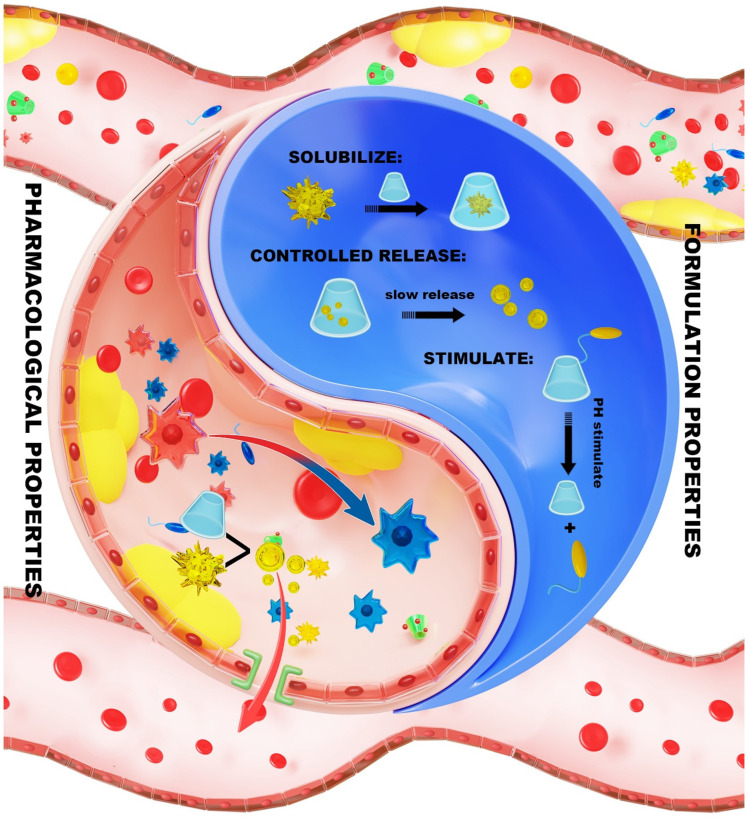
The dual functions of CDs in the treatment of AS. The figure was originally created by the authors using Adobe Photoshop 2020.

**Figure 6 pharmaceutics-17-01496-f006:**
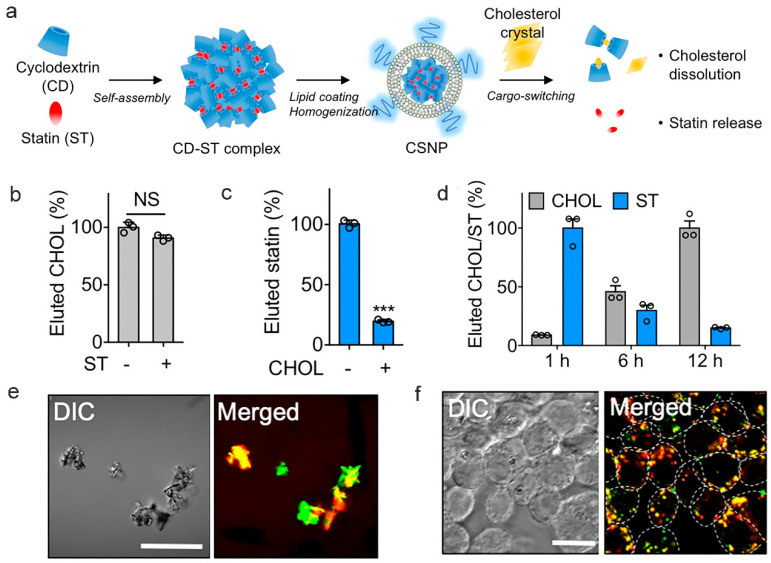
Affinity-Driven Design of Cargo-Switching Nanoparticles for AS Therapy [165]. (**a**): Schematic representation of cargo-switching nanoparticle (CSNP) preparation and cargo-switching mechanisms. (**b**,**c**): Competitive binding assays reveal that Chol exhibits a higher affinity for CSNPs than statin (ST). (**d**): Release of statin and dissolution of Chol is mediated by the CD–ST complexes through cargo-switching. (**e**): Confocal microscopy images demonstrate that CSNPs bind to Chol complexes (CCs) via cargo-switching interactionss. (**f**): Representative confocal microscopic images of CC-laden macrophages after CSNP treatment. Abbreviation guide, CSNP: Cyclodextrin-Statin Nanoparticles; CHOL: Cholesterol; DIC: Differential Interference Contrast; NS: Not Significant; Merged: Multi-channel fluorescence signal merged image. Statistical Significance Markers: In scientific research, *** indicates *p* < 0.001. Reprinted from Ref. [165], 2020, Kim, H. et al.

**Figure 7 pharmaceutics-17-01496-f007:**
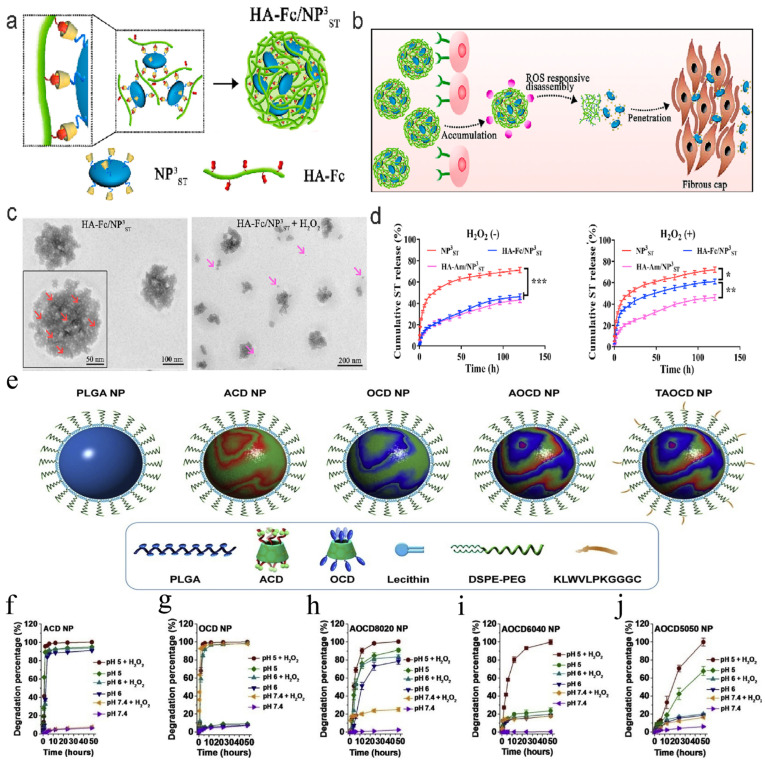
The ROS-responsive size-reducible HA-Fc/NP3 statin (ST) nanoassemblies and their intended performance [123]. (**a**): Small-sized NP3 ST is crosslinked by HA-Fc conjugates to form larger HA-Fc/NP3 ST nanoassemblies through multivalent host-guest recognition betweenβ-CD and Fc. (**b**): After accumulating in atherosclerotic plaques through HA-CD44 recognition, HA-Fc/NP3 ST rapidly disassembles in response to excess reactive oxygen species (ROS) in the intima, releasing smaller NP3 ST, which facilitates enhanced plaque penetration. (**c**): Transmission electron microscopy (TEM) images demonstrate that HA-Fc/NP3 ST degrades at high concentrations of H_2_O_2_. The red arrows indicated the discoidal particles in the TEM image of HA-Fc/NP3ST, and the purple arrows indicated the released NP3 ST with small size in the TEM image of HA-Fc/NP3 ST+H_2_O_2_. (**d**): The drug release rate demonstrates responsiveness to H_2_O_2_ stimulation. Statistical Significance Markers: In scientific research, * typically indicates *p* < 0.05, ** indicates *p* < 0.01, and *** indicates *p* < 0.001. Reprinted from Ref. [123]. Design and engineering of pH/ROS dual-responsive nanotherapies for targeted treatment of restenosis [136]. (**e**): Schematic illustration of different NPs examined in this study. (**f**–**j**): Hydrolysis curves of different NPs in PBS at pH 5, pH 6, or pH 7.4, with or without 1 mM H_2_O_2_. Reprinted from Ref. [136].

**Figure 8 pharmaceutics-17-01496-f008:**
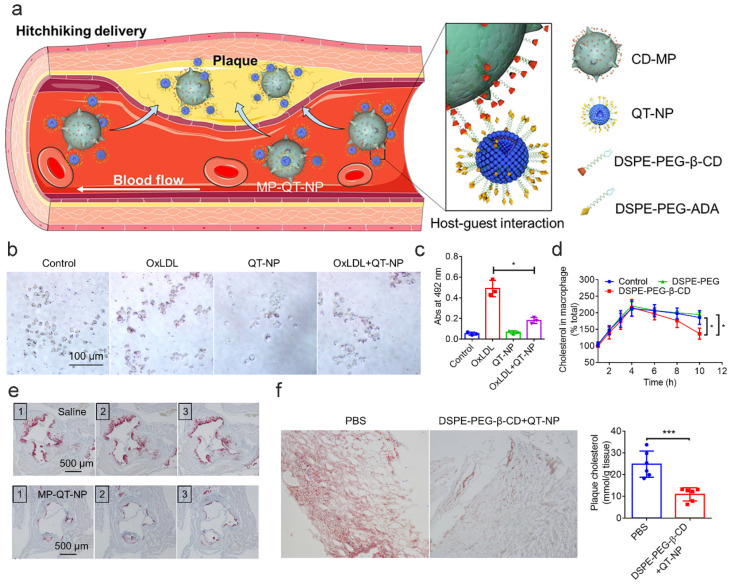
The CD-NDDS targeting CCs clearance through a hitchhiking strategy for the treatment of AS [143]. In (**a**), DSPE-PEG-β-CD modified macrophages are integrated with the QT-NP construct (MP-QT-NP) via the host–guest recognition between β-CD and adamantane. By leveraging the recruitment of macrophages during atherosclerotic progression, these macrophages facilitate the entry of liposomes into the targeted tissue, thereby creating an enhanced macrophage-hitchhiking delivery system that significantly increases liposome accumulation in the aortic lesions associated with AS. (**b**–**d**) show that in vitro experiments reveal QT-NP’s remarkable dissolution and clearance capabilities for CCs. (**e**) presents animal studies confirming the substantial CC clearance efficacy of QT-NP. Finally, panel (**f**) indicates that human carotid plaques treated with the combination of DSPE-PEG-β-CD and QT-NP exhibit reduced cholesterol content. Statistical Significance Markers: In scientific research, * typically indicates *p* < 0.05, *** indicates *p* < 0.001. These denote statistically significant differences between groups, with more asterisks indicating greater significance. Reprinted from Ref. [143], 2022, Gao, C. et al.

**Figure 9 pharmaceutics-17-01496-f009:**
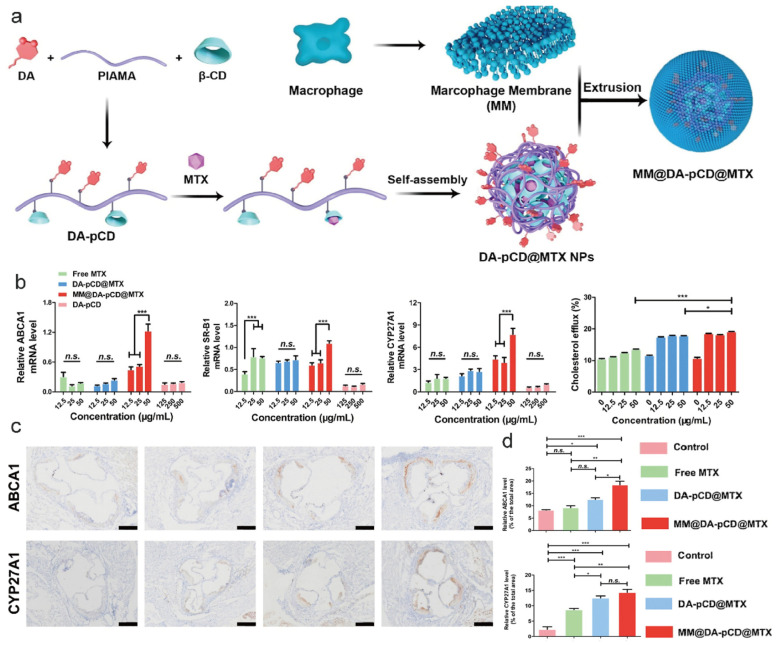
The biomimetic CD-NDDS (MM@DA-pCD@MTX) effectively enhances Chol efflux in treating AS [177]. (**a**): Illustration of the preparation process of the CD-NDDS, utilizing a self-assembly strategy integrated with cell membrane modification techniques. (**b**): Enzyme-linked immunosorbent assay (ELISA) results indicate that this biomimetic delivery system upregulates genes associated with Chol efflux, thereby facilitating Chol excretion. (**c**,**d**): Immunohistochemical analyses reveal that this delivery system promotes Chol efflux at the animal level, contributing to the mitigation of AS progression. Furthermore, it significantly upregulates the protein expression of ABCA1 and CYP27A1 within atherosclerotic plaques, thus accelerating cholesterol reverse transport and enhancing the functionality of transport proteins. The illustration of the preparation of MM@DA-pCD@MTX CD highlights its efficiency in encapsulating CCs, significantly increasing Chol solubility. Statistical Significance Markers: In scientific research, * typically indicates *p* < 0.05, ** indicates *p* < 0.01, and *** indicates *p* < 0.001. These denote statistically significant differences between groups, with more asterisks indicating greater significance. n.s.: meaning “no significant difference”. Reprinted from Ref. [177], 2024, Li, Z. et al.

**Figure 10 pharmaceutics-17-01496-f010:**
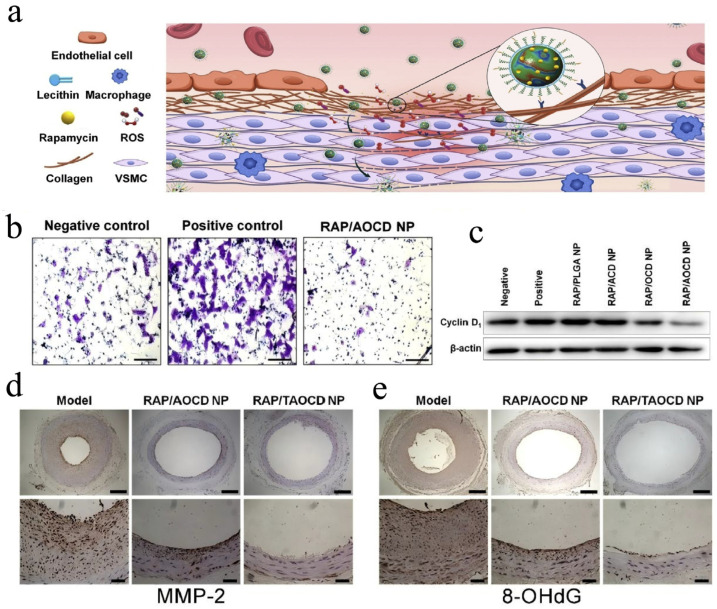
CD-NDDS significantly alleviates the inflammatory atherosclerotic microenvironment [136]. (**a**): The preparation of the pH/ROS dual-responsive CD-NDDS, composed of ACD/OCD, is illustrated. In vascular inflammatory microenvironments, these carriers respond to acidic pH and elevated levels of ROS, facilitating precise dissociation and drug release. (**b**,**c**): In vitro assays demonstrate that the CD-NDDS effectively inhibits the migration and proliferation of vascular smooth muscle cells induced by the inflammatory factor PDGF-BB, blocking aberrant cell proliferation through the downregulation of cyclin D1. (**d**,**e**): In vivo targeted treatments further confirm the carrier’s selective accumulation at lesion sites, significantly reducing markers of oxidative damage, such as 8-OHdG, and MMP-2. This interplay facilitates the reconstruction of the pathological microenvironment characterized by inflammation and an oxidative stress imbalance. Reprinted from Ref. [136].

**Table 1 pharmaceutics-17-01496-t001:** Summary of CD-NDDS for the Treatment of AS.

Types of CDs	Carrier Types	Nano-Systems	Drugs	Targeting Mechanisms	Animal Model	Route of Administration	Formulation Properties of CDs	Pharmacological Properties of CDs	References
HP-β-CD	Micelle	MTX NPs	Methotrexate (antimetabolite anticancer drug)	Host–guest recognition	Male Sprague-Dawley rats	Oral gavage	Enhanced drug solubility, improved encapsulation	Dissolve cholesterol crystals	[152]
β-CD	Nanogels	8-arm polyethylene glycol 20,000-CD	5-FU (antimetabolite chemotherapy); MTX (antimetabolite antitumor drug)		Prostate cancer model	tail Vein injection	Sustained release and high loading efficiency	Improve local drug concentration, enhance anticancer efficacy, reduce systemic toxicity	[68]
CD-NH_2_	Polymeric Nanoparticle	poly-β-CD/pH-sensitive benzimidazole-modified dextran sulfate/spherical nucleic acid		pH-responsive	ApoE^−^/^−^ mice (apolipoprotein E knockout mice)	Intravenous injection.	Stimuli-responsiveness, self-assembly stability, anti-hemolytic activity	Cholesterol-dissolving capacity, biocompatibility	[103]
β-CD	Polymeric Nanoparticle	PLGA-NPs	Atorvastatin (hepitorin)	HA	ApoE^−^/^−^ mice (apolipoprotein E gene knockout mice)	Intravenous injection	Enhanced drug solubility, sustained release	Enhance drug bioavailability, reduce systemic toxicity, and improve targeting efficiency	[153]
β-CD	Polymeric Nanoparticle	β-CD-CS	Broad-spectrum antibacterial agent	Chitosan-mediated biofilm adhesion			Enhanced drug solubility and loading capacity	Low toxicity and excellent biocompatibility	[154]
β-CD	Polymeric Nanoparticle	β-CD-PAA-PMMA	Broad-spectrum antibacterial agent	Surface charge-mediated targeting			sustained release	Good biocompatibility; low cytotoxicity	[154]
HP-β-CD	Polymeric Nanoparticle	CDNPs	Nile red (fluorescent dye), Indocyanine Green (near-infrared fluorescent dye)	(EPR) effect	ApoE^−^/^−^ mice (apolipoprotein E gene knockout mice)	tail Vein injection	Solubilization, stability, sustained release	Cholesterol-dissolving capacity, biocompatibility	[13]
β-CD	Liposome	HA-Fc/NP3 ST	Simvastatin (statins)	HA	ApoE-deficient mice (apoprotein E gene knockout mice)	Tail vein injection	Solubilization	ROS-responsive, promote deep plaque penetration	[123]
β-CD-NH_2_	Nanogels	shell-crosslinked nanoparticles (SCNPs)	Paclitaxel, Camptothecin (antitumor drug)	Host–guest recognition	HeLa cervical cancer xenograft model	Tail vein injection	Improved drug loading capacity and colloidal stability	Dissolve cholesterol crystals, confer redox responsiveness	[75]
β-CD	Inorganic Nanoparticle	MMSGNR-AlPcS4	AlPcS4 + Pt(IV) prodrug (a complex of photosensitizer and platinum prodrug)	Lactobionic acid targeting ligand	BALB/c nude mice with HepG2 human hepatocellular carcinoma cells	Tail vein injection	Controlled release; sealing mesoporous channels to prevent drug leakage	Reduction-responsive	[155]
β-CD	Polymeric Nanoparticle	AOCD NP, TAOCD NP	Rapamycin (mTOR inhibitor)	(EPR) effect	Carotid artery balloon injury induced vascular inflammation rats	Intravenous injection	Dual-responsive carrier materials: pH-sensitive (ACD component), ROS-sensitive (OCD component)	Improve drug-loading capacity	[136]
PH-CD, HA-CD	Polymeric Nanoparticle	dual-carrier nanoparticles (Double-NPs)	Epicatechin gallate (flavonoid compound)	HA	Superscale ApoE^−^/^−^ mice (apolipoprotein E gene knockout mice)	Intraperitoneal injection	Enhanced drug stability, dual-carrier co-delivery, and low pH responsiveness	Alleviate inflammatory microenvironment; in vivo targeting of atherosclerotic plaques	[113]
β-CD	Micelle	MM@MTX NPs	MTX (immunosuppressant)	Cell membrane	ApoE^−^/^−^ mice (apolipoprotein E gene knockout mice)	Tail vein injection	Host–guest recognition; controlled release	Promote cholesterol dissolution and efflux, inhibit foam cell formation	[102]
α-CD	Micelle	Poly(ethylene glycol)-polylactic acid	DOX (antitumor drug)	(EPR) effect			Controlled release	Solubilization-promoting, prolongs drug circulation time in vivo	[156]
β-CD	Nanogels	β-CD/Polyvinyl alcohol-co-poly(2-acrylamide-2-methylpropane sulfonic acid) cross-linked hybrid Interpenetrating polymer networks-nanogels	Rosuvastatin calcium, RST (statins)	(EPR) effect	New Zealand White Rabbit with high blood fat (1400–1500 g)	Oral administration	Improved stability; solubilization	Promote absorption; dissolve cholesterol	[128]

## Data Availability

No new data was created or analyzed in this study.

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
