# Peer review of "Cyclodextrin: Dual Functions as a Therapeutic Agent and Nanocarrier for Regulating Cholesterol Homeostasis in Atherosclerosis"

_pharmaceutics, 2025, doi:10.3390/pharmaceutics17111496_

Round 1
Reviewer 1 Report
Comments and Suggestions for Authors
Manuscript entitled " Cyclodextrin-Based Nanomedicine: Dual-Functional Carrier and Therapeutic Agent for Targeted Cholesterol Homeostasis Regulation in Atherosclerosis " by Cui et al.
The authors discuss the problem of atherosclerosis in relation to imbalances in cholesterol
metabolism. The main attention is paid to the cyclodextrins-based nano drug delivery systems that can effectively solubilize cholesterol crystals, penetrate plaque barriers and deliver active agents.
The introduction provides sufficient to clearly state the discussed problem. Importantly, most of the references refer to the recent years that underline the actuality and timeliness of the review.
The types and structures of nano drug delivery systems based on CDs for atherosclerosis treatment are described comprehensively and the most prevalent are marked.
A separate section "Challenges and Perspectives" contains the challenges of using cyclodextrins depending on their structure and suggest methods to increase safety of CDs with enzyme-sensitive polymeric cyclodextrins or using CDs with negatively charged groups.
Interesting part describes that cholesterol imbalance not only causes atherosclerosis, but also it is associated with tumor growth. Extracting cholesterol from membranes inhibits tumor cell proliferation and metastasis.
List of abbreviations is useful to understand the short names of the terms used.
The review can be accepted for publication after addressing the following minor comments:
- Make punctuation uniform throughout the manuscript: for ex.
Line 42 "……lipids and forming foam cells.[3,4]." But
Line 45 "……… oxidative stress responses[5]."
Line 530
and so forth…
- Line 64
"…plaque barriers and deliver active agents and release drugs …" replace with:
"…plaque barriers, deliver active agents and release drugs …"
- Line 373
Inorganic/Metal Hybrid CD Systems
It is interesting if these systems have common features with the Metal Organic Frameworks based on cyclodextrins?
It would be useful to address the issues of using CD NOF for atherosclerosis therapy and cardiovascular diseases (Li et al. 2025 https://doi.org/10.1016/j.bioactmat.2024.08.041, Ge et al. 2025 https://doi.org/10.1016/j.ccr.2025.216971)
- Line 518
"…and liposomes[120]Substantial…." Should be "…and liposomes[120]. Substantial…."
Lines 830-836
"…adding negatively charged groups… " As far as it is known, SBE-β-CD has negatively charged sulfobutyl ether groups. What about the safe doses of this CD?
Author Response
Title: Cyclodextrin: dual functions as a therapeutic agent and nanocarrier for regulating cholesterol homeostasis in atherosclerosis
Dear Editor:
We feel great thanks for your professional review work on our article. As you are concerned, there are several problems that need to be addressed. Our response is given in normal font and changes/additions to the manuscript are given in the red text. We hope that the revised manuscript will meet the publication requirements of the journal.
Reviewer #1:
- The authors discuss the problem of atherosclerosis in relation to imbalances in cholesterol metabolism. The main attention is paid to the cyclodextrins-based nano drug delivery systems that can effectively solubilize cholesterol crystals, penetrate
The review can be accepted for publication after addressing the following minor comments:
Make punctuation uniform throughout the manuscript: for ex.
Line 42 "……lipids and forming foam cells.[3,4]." But
Line 45 "……… oxidative stress responses[5]."
Line 530
and so forth…
Response:We sincerely appreciate your suggestions. We have implemented all the suggested revisions you provided within the manuscript. As other reviewers' comments necessitated modifications to the original text, the specific lines you referenced Line 42 and Line 45 do not appear in their original locations. However, we have systematically replaced all relevant instances with the standardized format "such as SBE-β-CD[199].".
2.Line 64"…plaque barriers and deliver active agents and release drugs …" replace with:"…plaque barriers, deliver active agents and release drugs …"
Response: We sincerely appreciate your suggestions. We have implemented the suggested revisions with tracked changes in Page 2, Lines 61–62.
3.Line 373,It is interesting if these systems have common features with the Metal Organic Frameworks based on cyclodextrins?
It would be useful to address the issues of using CD NOF for atherosclerosis therapy and cardiovascular diseases (Li et al. 2025 https://doi.org/10.1016/j.bioactmat.2024.08.041, Ge et al. 2025 https://doi.org/10.1016/j.ccr.2025.216971)
Response: We greatly appreciate your suggestions. Regarding the points you raised, we have addressed them in page 10, lines 385-413, with the relevant sections highlighted in red.
4.Line 518,"…and liposomes[120]Substantial…." Should be "…and liposomes[120]. Substantial…."
Response:We greatly appreciate your suggestions. Regarding the feedback you provided, we have made changes in page 18, lines 532-534. Since the content you requested to be altered was an example, we determined that this example was not appropriate within the text and therefore removed it. Consequently, the content you referenced does not appear in the revised version.
5.Lines 830-836,"…adding negatively charged groups… " As far as it is known, SBE-β-CD has negatively charged sulfobutyl ether groups. What about the safe doses of this CD?
Response: Thank you for your careful review of our manuscript and for your valuable comments. Regarding the safety dosage of SBE-β-CD, a cyclodextrin with negatively charged sulfobutylether groups, based on current literature data, a specific safety value applicable to all scenarios cannot be provided. This is because it requires detailed toxicological and pharmacokinetic studies tailored to the specific drug, administration route, target disease, and patient conditions to determine.
Safety evaluation of nanodrug delivery systems is a complex process, typically involving comprehensive analysis of the nanomaterial composition, size, surface chemistry, encapsulated entities, and potential immunogenicity[1]. For nanoparticles and nanodrug delivery systems, studies characterize their morphology and size using various methods such as scanning electron microscopy, transmission electron microscopy, and dynamic light scattering. Meanwhile, drug release kinetics and biological analysis are also crucial to evaluate their in vivo behavior and potential immune responses. For example, release studies on cyclin-dependent kinase inhibitor AS2863619, TGF-β, and IL-2 have shown that the release percentage of these drugs reaches approximately 66% within 120 hours[2].
In summary, SBE-β-CD exhibits broad application prospects in drug delivery due to its unique structure and excellent physicochemical properties, with lower nephrotoxicity compared to other cyclodextrin derivatives. However, determining its safe dosage in specific clinical applications requires extensive and systematic pharmacological and toxicological studies, combined with clinical trial data for comprehensive evaluation. Currently, a universal specific safety dosage value for SBE-β-CD cannot be provided.
Accordingly, we have revised the main text, as seen in page 27, line 825-827.

Reviewer 2 Report
Comments and Suggestions for Authors
In this manuscript, the authors describe "Cyclodextrin-Based Nanomedicine: Dual-Functional Carrier and Therapeutic Agent for Targeted Cholesterol Homeostasis Regulation in Atherosclerosis" Although this manuscript is well written, it would be helpful if the authors address the concerns below.
1) The entire manuscript is rife with many unnecessary acronyms some of which includes Chol, CD, CC, AS, CD-NDDS...etc. The excessive use of many acronyms in this manuscript has the potential to distract potential readers and even cause confusion as well. It would be very helpful if the authors revise these by significantly reducing the use of acronyms in the manuscript.
2) In line 731 there was no reference cited for Zhu et al. The authors should include that.
In summary, this manuscript could benefit its target readers is the above concerns are addressed.
Author Response
Title: Cyclodextrin: dual functions as a therapeutic agent and nanocarrier for regulating cholesterol homeostasis in atherosclerosis
Dear Editor:
We feel great thanks for your professional review work on our article. As you are concerned, there are several problems that need to be addressed. Our response is given in normal font and changes/additions to the manuscript are given in the red text. We hope that the revised manuscript will meet the publication requirements of the journal.
Reviewer #2:
- The entire manuscript is rife with many unnecessary acronyms some of which includes Chol, CD, CC, AS, CD-NDDS...etc. The excessive use of many acronyms in this manuscript has the potential to distract potential readers and even cause confusion as well. It would be very helpful if the authors revise these by significantly reducing the use of acronyms in the manuscript.
- In line 731 there was no reference cited for Zhu et al. The authors should include that.
Response: We greatly appreciate your suggestion and have incorporated it in page 23, line 729.

Reviewer 3 Report
Comments and Suggestions for Authors
Dear authors, the thematic of this review is quite interesting and it fits with the aims and scope of the Journal of Pharmaceutics. However a major revision is need to increased the readability by scientists. The title should be better focused, it is not immediately clear the meaning. i.e. Cyclodextrins: dual function as therapeutic agents and nanocarriers for.... etc...
Main suggestion is to short the manuscript, i.e. only the essential parts should be introduced. Concerning CD as nanocarriers please only describe those are related with their use in the case of hypercholesterolaemia.
Table 1 : it is essential the authors separate the publications related to CD as therapeutic agents from the references related to CD as nanocarries or supramolecular complexes loading therapeutic agents such as statins. The route of administration is also very important to be added in the table together with the animal model used or if it was tested or currently used in humans.
Also the loaded drugs are very different, please separate the different classes.
Author Response
Title: Cyclodextrin: dual functions as a therapeutic agent and nanocarrier for regulating cholesterol homeostasis in atherosclerosis
Dear Editor:
We feel great thanks for your professional review work on our article. As you are concerned, there are several problems that need to be addressed. Our response is given in normal font and changes/additions to the manuscript are given in the red text. We hope that the revised manuscript will meet the publication requirements of the journal.
Reviewer #3:
1.Dear authors, the thematic of this review is quite interesting and it fits with the aims and scope of the Journal of Pharmaceutics. However a major revision is need to increased the readability by scientists. The title should be better focused, it is not immediately clear the meaning. i.e. Cyclodextrins: dual function as therapeutic agents and nanocarriers for.... etc...
Response: Thank you very much for your suggestion. The title have been better focused in Page 1, Lines 2–4.
- Main suggestion is to short the manuscript, i.e. only the essential parts should be introduced. Concerning CD as nanocarriers please only describe those are related with their use in the case of hypercholesterolaemia.
3.Table 1 : it is essential the authors separate the publications related to CD as therapeutic agents from the references related to CD as nanocarries or supramolecular complexes loading therapeutic agents such as statins. The route of administration is also very important to be added in the table together with the animal model used or if it was tested or currently used in humans. Also the loaded drugs are very different, please separate the different classes.
Response: Thank you very much for your suggestion. First, we have already distinguished between “Formulation properties of CDs” and “Pharmacological properties of CDs” in the table. The original intent was to differentiate whether cyclodextrins function as therapeutic agents or as supramolecular complexes for drug loading, etc.Second, we have added two supplementary columns to Table 1: Animal model and Route of administration. Furthermore, in Table 1, the Drugs column distinguishes the loaded medications. Finally, none of the animal models cited in the paper have undergone human trials.

Reviewer 4 Report
Comments and Suggestions for Authors
This review article provides a comprehensive overview of the development and application of cyclodextrin-based nano-drug delivery systems (CD-NDDS) for the treatment of atherosclerosis (AS). The authors position CD not only as a versatile nanocarrier but also as an active therapeutic agent capable of regulating cholesterol homeostasis by dissolving cholesterol crystals, promoting cholesterol efflux, and alleviating inflammatory microenvironments. The manuscript is well-structured, systematically covering the pathological basis of AS, the molecular interactions of cyclodextrins with cholesterol, construction strategies of various nanoplatforms, and their functional applications. A significant strength is the integration of formulation properties (solubilization, controlled release, stimulus response) with pharmacological actions (cholesterol dissolution, efflux promotion, anti-inflammation), supported by numerous recent examples and clear schematics.
The review is highly comprehensive and highly relevant to the field of nanomedicine and cardiovascular therapeutics. It successfully identifies a clear gap: the limitations of current lipid-lowering therapies in addressing the core pathological issues of cholesterol crystal deposition and the associated inflammatory microenvironment in advanced plaques. The focus on CD's dual functionality as both a carrier and a therapeutic agent is a compelling and modern perspective that distinguishes this review.
While there have been reviews on cyclodextrins in drug delivery and some on nanomedicine for atherosclerosis, this manuscript offers a unique and focused synthesis on the specific role of CD-based systems for targeted cholesterol homeostasis regulation. Its detailed analysis of structure-activity relationships, diverse construction strategies (micelles, polymers, liposomes, nanogels, inorganic hybrids), and the mechanistic breakdown of CD's pharmacological properties provides significant added value beyond existing literature.
The reference list is extensive and largely appropriate, incorporating a good mix of foundational knowledge and recent (within the last 5 years) high-impact research. The citations effectively support the claims made throughout the text. There is no apparent excessive self-citation.
The statements and conclusions are generally coherent and well-supported by the cited literature. The logical flow from the pathology of AS to the solution offered by CD-NDDS is clear and persuasive.
The conceptual figures (e.g., Fig. 3, 5, 6, 7, 8, 9, 10) are appropriately designed and greatly aid in understanding the complex nanoplatforms and mechanisms. Table 1 is a valuable summary of the various CD-NDDS.
Minor Points
1. Section 8 (Challenges and Perspectives) is critical and well-started. However, the discussion on biocompatibility (Lines 818-830) could be enhanced by proposing more specific molecular engineering solutions beyond the general "enzyme-sensitive poly-CD" or "adding negatively charged groups". Citing specific examples from the literature where such strategies have successfully reduced cytotoxicity would be more impactful.
2. Table 1: This is a very useful summary. For greater clarity, consider adding a column for the key in vivo model used (e.g., ApoE-/- mice) or the primary demonstrated outcome (e.g., "Reduced plaque area", "Increased cholesterol efflux") to provide a quick comparative overview of the efficacy of the different systems.
Author Response
Title: Cyclodextrin: dual functions as a therapeutic agent and nanocarrier for regulating cholesterol homeostasis in atherosclerosis
Dear Editor:
We feel great thanks for your professional review work on our article. As you are concerned, there are several problems that need to be addressed. Our response is given in normal font and changes/additions to the manuscript are given in the red text. We hope that the revised manuscript will meet the publication requirements of the journal.
Reviewer #4:
- Section 8 (Challenges and Perspectives) is critical and well-started. However, the discussion on biocompatibility (Lines 818-830) could be enhanced by proposing more specific molecular engineering solutions beyond the general "enzyme-sensitive poly-CD" or "adding negatively charged groups". Citing specific examples from the literature where such strategies have successfully reduced cytotoxicity would be more impactful.
- Table 1: This is a very useful summary. For greater clarity, consider adding a column for the key in vivo model used (e.g., ApoE-/- mice) or the primary demonstrated outcome (e.g., "Reduced plaque area", "Increased cholesterol efflux") to provide a quick comparative overview of the efficacy of the different systems.
Response: We greatly appreciate your suggestion and have added a column titled “Animal model” to Table 1 to address it, thereby enhancing the completeness of the manuscript. Due to space constraints, we have chosen to include only the animal model information in the table, and the main results are not elaborated upon in detail within the table itself.

Reviewer 5 Report
Comments and Suggestions for Authors
Hao Cui et al., have presented the draft on Cyclodextrin-Based Nanomedicine: Dual-Functional Carrier and Therapeutic Agent for Targeted Cholesterol Homeostasis Regulation in Atherosclerosis. Here are some of the comments for the same.
- Specific synergistic pathways like efflux via ABCA1, inflammasome inhibition are missing in synergistic functions. (Line 20–23).
- In the introduction, the role of cholesterol crystals in plaque rupture is described. Please add more mechanistic insight on inflammasome activation and pathways involved into the same. (Line 42–48).
- In description of CD as a supramolecular compound, include more recent structural studies confirming guest–host interactions with cholesterol. (Line 59–66).
- The cholesterol metabolism section is descriptive but lacks a schematic summarizing key pathways for biosynthesis, transport, efflux through diagrammatic representation (Line 90–95).
- More recent reviews/research (2023–2024) on cholesterol homeostasis and AS should be added. (Line 130–136).
- The description of host–guest micellar interactions is detailed, but it is unclear how stable these systems are under physiological ionic strengths. Justify the same (Line 222–236).
- Covalent bonding strategies are introduced but examples are limited to CD–epichlorohydrin. Please elaborate the click chemistry approach for the same. (Line 237–245).
- Figure 2 is introduced but lacks sufficient caption detail to be self-explanatory.
- The role of PEGylated CDs is explained; however, pharmacokinetics including clearance times, RES uptake are not compared. (Line 300–305).
- The inflammatory microenvironment discussion does not adequately address macrophage polarization (M1/M2). Please justify the same. (Line 350–356).
- The cell membrane coating strategy is promising, but practical challenges like batch reproducibility, immune clearance should be critically discussed. (Line 452–462).
- The gene pathway regulation by CDs is described (ABCA1, CYP27A1), but lacks mention of transcriptomic/proteomic confirmation. Suggest including recent omics studies. (Line 725–736).
- The inflammatory microenvironment discussion is strong, but vascular smooth muscle cell effects need clearer linkage to AS pathology. (Line 793–803).
- Challenges section should mention long-term toxicity of HP-β-CD notably ototoxicity and renal concerns reported in vivo. (Line 821–829).
- Regulatory considerations (EMA/FDA approval of CD-based excipients) are missing. (Line 850–856).
- Future perspectives are generic; The data on AI-driven CD design, microfluidic scaling are missing. (Line 893–898).
Author Response
Title: Cyclodextrin: dual functions as a therapeutic agent and nanocarrier for regulating cholesterol homeostasis in atherosclerosis
Dear Editor:
We feel great thanks for your professional review work on our article. As you are concerned, there are several problems that need to be addressed. Our response is given in normal font and changes/additions to the manuscript are given in the red text. We hope that the revised manuscript will meet the publication requirements of the journal.
Reviewer #5:
1.Specific synergistic pathways like efflux via ABCA1, inflammasome inhibition are missing in synergistic functions. (Line 20–23).
Response: Thank you very much for your suggestion. We have made the revision on Page 1, Lines 21–23 of the document.
2.In the introduction, the role of cholesterol crystals in plaque rupture is described. Please add more mechanistic insight on inflammasome activation and pathways involved into the same. (Line 42–48).
Response: Thank you very much for your suggestion. We have made the revision on Page 1, Lines 40–58 of the document.
3.In description of CD as a supramolecular compound, include more recent structural studies confirming guest–host interactions with cholesterol. (Line 59–66).
Response: Thank you very much for your suggestion. We have made the revision on Page 2, Lines 51–58 of the document.
4.The cholesterol metabolism section is descriptive but lacks a schematic summarizing key pathways for biosynthesis, transport, efflux through diagrammatic representation (Line 90–95).
Response: Thank you very much for your suggestion. We have already described the synthesis and transport of key steps in Figure 1 on Page 3 of the manuscript.
5.More recent reviews/research (2023–2024) on cholesterol homeostasis and AS should be added. (Line 130–136).
Response: Thank you very much for your suggestion. We have made the revision on Page 4, Lines 131–139 of the document.
6.The description of host–guest micellar interactions is detailed, but it is unclear how stable these systems are under physiological ionic strengths. Justify the same (Line 222–236).
Response: Thank you very much for your suggestion. We have made the revision on Page 6, Lines 218–224 of the document.
7.Covalent bonding strategies are introduced but examples are limited to CD–epichlorohydrin. Please elaborate the click chemistry approach for the same. (Line 237–245).
Response: Thank you very much for your suggestion. We have made the revision on Page 6, Lines 245–252 of the document.
8.Figure 2 is introduced but lacks sufficient caption detail to be self-explanatory.
Response: Thank you very much for your suggestion. We have made the revision on Page 7, Lines 260–273 of the document.
9.The role of PEGylated CDs is explained; however, pharmacokinetics including clearance times, RES uptake are not compared. (Line 300–305).
Response: Thank you very much for your suggestion. We have made the revision on Page 8, Lines 313–316 of the document.
10.The inflammatory microenvironment discussion does not adequately address macrophage polarization (M1/M2). Please justify the same. (Line 350–356).
Response: Thank you very much for your suggestion. We have made the revision on Page 25-26, Lines 763–794 of the document.
11.The cell membrane coating strategy is promising, but practical challenges like batch reproducibility, immune clearance should be critically discussed. (Line 452–462).
Response: Thank you very much for your suggestion. We have made the revision on Page12, Lines 477–485 of the document.
12.The gene pathway regulation by CDs is described (ABCA1, CYP27A1), but lacks mention of transcriptomic/proteomic confirmation. Suggest including recent omics studies. (Line 725–736).
Response: Thank you very much for your suggestion. We have made the revision on Page4-Page5, Lines 157-Lines 173 of the document.
13.The inflammatory microenvironment discussion is strong, but vascular smooth muscle cell effects need clearer linkage to AS pathology. (Line 793–803).
Response: Thank you very much for your suggestion. We have made the revision on Page26, Lines 788–791 of the document.
14.Challenges section should mention long-term toxicity of HP-β-CD notably ototoxicity and renal concerns reported in vivo. (Line 821–829).
Response: Thank you very much for your suggestion. We have made the revision on Page27, Lines 815–827 of the document.
15.Regulatory considerations (EMA/FDA approval of CD-based excipients) are missing. (Line 828–843).
Response: Thank you very much for your suggestion. We have made the revision on Page27, Lines 851–867 of the document.
16.Future perspectives are generic; The data on AI-driven CD design, microfluidic scaling are missing. (Line 893–898).
Response: Thank you very much for your suggestion. We have made the revision on Page27, Lines 838–839, and Page28, Lines 868–876 of the document.

Round 2
Reviewer 3 Report
Comments and Suggestions for Authors
The revised manuscript is now acceptable for the publication
Author Response
We feel great thanks for your professional review work on our article. As you are concerned, there are several problems that need to be addressed. Our response is given in normal font and changes/additions to the manuscript are given in the red text. We hope that the revised manuscript will meet the publication requirements of the journal.
Reviewer 5 Report
Comments and Suggestions for Authors
- Plagiarism is high for revised version of the draft. It should be <20%.
- The explanation of host–guest micellar stability under physiological ionic strength is still largely qualitative. The section would benefit from inclusion of quantitative thermodynamic or molecular dynamics data validating the retention of host–guest complexes in saline or serum-like environments. (Lines 218–224)
- The pharmacokinetic discussion of PEGylated cyclodextrin-based carriers remains superficial. It is recommended to incorporate a concise summary or comparative table showing half-life, biodistribution, and reticuloendothelial system (RES) uptake between PEGylated and non-PEGylated CD formulations. (Lines 313–316)
- Although the manuscript elaborates on ABCA1 and CYP27A1 pathways, transcriptomic or proteomic confirmation of these mechanisms is missing. The authors should reference recent omics datasets or GEO accession studies validating upregulation of these genes following cyclodextrin exposure. (Lines 157–173)
- The section mentioning AI-driven microfluidic integration is conceptually strong but not critically discussed. The authors should include a schematic or descriptive workflow linking artificial intelligence-based molecular modeling with microfluidic synthesis optimization of CD nanocarriers. (Lines 1254–1258)
- The phrase “precise dissolution and metabolic remodeling of cholesterol crystals” appears repeated in the concluding discussion. (Lines 1248–1251)
- The legend of Figure 4 contains excessive procedural description and inconsistent labeling of subfigures (a–f). It should be simplified, with each subpanel clearly identified and correlated with its corresponding image for better readability and interpretability. (Lines 699–705)
- The manuscript still contains visible formatting residues such as “Formatted: Font color: Red,” duplicated sections, and irregular spacing or punctuation. All such editing artifacts should be completely removed to meet journal formatting and readability standards. (Lines 1293–1303; 1643–1651)
- The repeated concatenation of abbreviations without spacing—examples include “CyclodextrinCD” and “AtherosclerosisAS”—needs correction to the standardized forms “Cyclodextrin (CD)” and “Atherosclerosis (AS)” throughout the document. (Lines 39–131; 384–393)
- The description of dual-responsive (pH/ROS) nanocarriers would benefit from a schematic illustrating the drug release mechanism under acidic and oxidative conditions. (Lines 1213–1215)
- The abbreviation list at the end contains redundant or undefined terms (e.g., NP³â‚›â‚œ, LFP). It should be reviewed for internal consistency, ensuring each abbreviation is defined upon first mention and corresponds to usage within the main text. (Lines 1275–1292)
Comments on the Quality of English Language
The current version of the draft requires revision.
